# Disentanglement by means of action-induced representations

## Abstract

Learning interpretable representations with variational autoencoders (VAEs) is a major goal of representation learning. The main challenge lies in obtaining disentangled representations, where each latent dimension corresponds to a distinct generative factor. This difficulty is fundamentally tied to the inability to perform nonlinear independent component analysis. Here, we introduce the framework of action-induced representations (AIRs) which models representations of physical systems given experiments (or actions) that can be performed on them. We show that, in this framework, minimal AIRs provably disentangle degrees of freedom w.r.t. their action dependence. We further introduce a variational AIR architecture (VAIR) that systematically approximates such minimal AIRs and therefore can achieve disentanglement where standard VAEs fail. Beyond state representation, VAIR also captures the action dependence of the underlying generative factors, directly linking experiments to the degrees of freedom they influence.

Creating compressed representations of high-dimensional observations has become essential across a range of machine learning tasks, from improving computational efficiency in downstream pipelines (Rombach et al., 2022; Laskin et al., 2020), to enabling interpretable approaches that aim to uncover structure within the input data. In this context, variational autoencoders (VAEs) (Kingma & Welling, 2013) have emerged as a hallmark architecture for unsupervised representation learning. VAEs aim to compress input data into a low-dimensional latent space, creating a minimal representation of it. In cases where a dataset is characterized by few hidden variables, also known as factors of variation, VAEs are expected to extract these by encoding them in their latent space (Higgins et al., 2017). In physics, this translates to identifying the key features of a system, typically associated with its effective degrees of freedom (Iten et al., 2020; Kalinin et al., 2021; Nautrup et al., 2022; Valleti et al., 2024; Møller et al., 2025) or order parameters (Wetzel, 2017; de Schoulepnikoff et al., 2025).

However, several challenges remain. One that has attracted significant attention, due to its foundational importance, is the disentangled nature of the learned representations (Locatello et al., 2019). This refers to a representation in which each neuron in the VAE's latent space encodes a single factor of variation, in contrast to an entangled representation, where the factors are mixed within the latent neurons. Indeed, such a problem is closely related to independent component analysis (ICA) (Hyvarinen & Morioka, 2016), which aims to recover the underlying components from observed data. This connection has revealed that, under typical conditions where the mapping from factors of variation to observations is nonlinear, recovering the true generative factors becomes fundamentally ill-posed, due to the undecidable nature of nonlinear ICA (Khemakhem et al., 2020). From a physics standpoint, this implies that the VAE may not uncover the system's actual degrees of freedom, but instead learn entangled combinations of them, thus hindering the effective interpretability of the model. Indeed, substantial efforts have been made to address the issue, and numerous VAE variants have been proposed to mitigate it (Chen et al., 2018; Hsu et al., 2023; Kim & Mnih, 2018).

From a broader perspective, VAEs and their more advanced variants are typically trained on a fixed dataset of observations. This is analogous to analyzing a collection of experimental measurements. However, in scientific practice, the process by which data is acquired is often as important as the data itself. In experimental settings, one commonly probes a system by applying a perturbation, or action, and observing the system's

response. The nature of these actions thus encodes essential information that can provide deeper insight into the physical system under investigation. For example, certain actions may selectively couple to specific degrees of freedom, such as heating an object affecting its temperature, but not its mass.

In this work, we show that incorporating actions is key for learning disentangled representations, and we introduce a framework specifically designed to extract physical concepts in experimental settings. To this end, we develop a novel VAE variant, drawing inspiration from recent advances in representation learning within reinforcement learning and causal inference. The model is trained on pairs consisting of an action applied to the system and the resulting observation or measurement. The proposed architecture is built with two main objectives: first, to enforce the emergence of disentangled representations of the observed data; and second, to explicitly link each action to the specific degrees of freedom it affects.

We begin by presenting related works in Section 1. Then, in Section 2 we present the theoretical analysis of the framework. We describe how and why disentanglement arises, and identify the conditions under which a set of actions induces a disentangled latent space. We then present in Section 3 a VAE architecture capable of producing such disentangled representations. Last, in Section 4 we numerically demonstrate the model's capabilities in three experiments: an abstract example designed to benchmark the capabilities of the model and show its advantage in disentanglement w.r.t. state-of-the-art VAE approaches, and two physics-inspired scenarios: one involving the recovery of the degrees of freedom of a classical particle from its trajectory, and another concerning the representation of small quantum systems from tomographic data.

# 1 Related work

**VAE-based disentanglement.** Variational autoencoders (Kingma & Welling, 2013) have become a cornerstone of unsupervised representation learning, with a large body of work dedicated to improving the disentanglement of their latent representations. $\beta$-VAE (Higgins et al., 2017) introduced a weighted KL regularization to encourage independence among latent neurons, while subsequent works proposed alternative objectives such as the Total Correlation penalty (Chen et al., 2018) and the FactorVAE loss (Kim & Mnih, 2018). Despite these efforts, Locatello et al. (2019) showed that unsupervised disentanglement is fundamentally ill-posed without inductive biases or additional supervision, a result that directly motivates our action-based approach. More recently, disentanglement via latent quantization (Hsu et al., 2023) and other structural priors have been proposed, but none of these methods exploit the structure of experimental interventions as a source of disentanglement, which is the key contribution of our work.

**Nonlinear ICA and identifiability.** The connection between disentanglement and independent component analysis (ICA) has been extensively studied. While linear ICA is well understood, nonlinear ICA is fundamentally ill-posed without additional structure (Hyvarinen & Morioka, 2016), as the underlying sources cannot be uniquely recovered from observations alone. Khemakhem et al. (2020) showed that identifiability can be restored by conditioning on auxiliary variables, leading to the iVAE framework. Our work draws inspiration from this insight, but differs in a key way: rather than modifying the prior distribution, we use actions to control which latent variables are active for a given observation, achieving provable disentanglement through the intersection structure of the action set.

**Causal and multi-view representation learning.** A closely related line of work is causal representation learning (Schölkopf et al., 2021), which seeks to recover hidden causal variables through interventions. In this setting, actions or interventions are typically assumed to modify the hidden causal variables themselves, and algorithms are designed to infer these transformations (Lachapelle et al., 2024; Gendron et al., 2023). Our framework operates under a fundamentally different paradigm: first, it does not consider a causal graph nor causal relation between latent factors; second, the intrinsic properties of the system remain fixed, and actions yield different observations that reflect them. This distinction makes our approach particularly well-suited to physical and experimental settings, where the system's degrees of freedom are stable properties that are probed rather than altered by experiments (see Figure 1a). Interestingly, a recent line of multi-view and multimodal work studies an identifiability statement related to Theorem 1, in which the recoverable latents are those shared across the subsets of variables that different views or modalities depend on. For instance, Yao et al. (2024) recover the latents common to several views as the intersection of what each view sees,

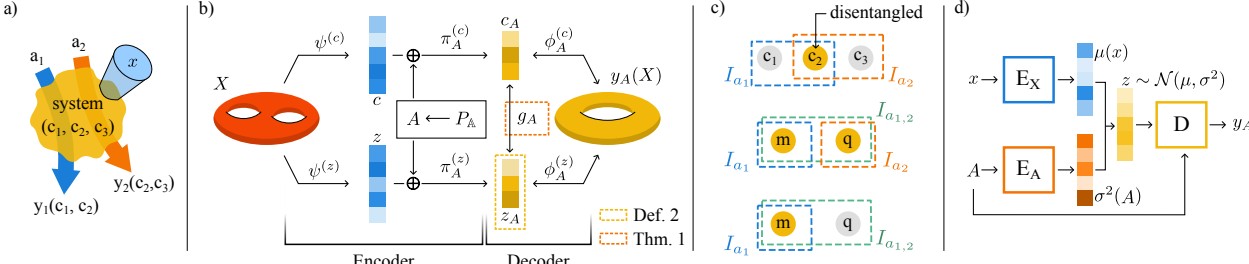

Figure 1: **a)** Schematic representation of the problem considered. A physical system, uniquely described by its factors of variation $\mathbf{c}$, is observed in a certain state $x$. Some elementary actions $a_i \in \mathbb{A}$ are applied to the system, leading to outcomes $y_i(\mathbf{c})$ that may only depend on a subset of $\mathbf{c}$. **b)** Diagrammatic aid of the theoretical representation of the problem. See main text for details. **c)** Top scheme represents the factors of variation $\mathbf{c}$ of a), together with the subsets $I_A$ corresponding to each action $a_i$. Because $c_2$ is required for both actions, it is in the intersection $I_{a_1} \cap I_{a_2}$ and by Theorem 1 will be disentangled in any minAIR (yellow color). The two lower schemes represent the same picture, here for Example 3 (main text), and for two different action combination sets $P_{\mathbb{A}} = \{\{1\}, \{2\}, \{1, 2\}\}$ and $P_{\mathbb{A}} = \{\{1\}, \{1, 2\}\}$, respectively. **d)** VAIR: two encoders ($E_X$ and $E_A$) compute the latent neurons' mean and variance from an input sample $x$ and action combination $A$ (or elementary action $a$), respectively. A latent sample $z$ and the $A$ are then fed into the decoder $D$ to compute the output $y_A$.

with rules for which groups are identifiable, while Daunhawer et al. (2023) and Sturma et al. (2023) treat the two-modality and unpaired multi-domain cases. These works recover shared latents from views co-observed on the same sample, whereas AIR obtains an analogous intersection structure from the action-dependence of a fixed physical system, through the deterministic minimality of Theorem 1 rather than a contrastive objective between views.

**Representation learning with actions and interactions.** The idea of using interactions with an environment to learn structured representations has been explored in the context of reinforcement learning. For instance, (Thomas et al., 2018; Sawada, 2018) proposed to disentangle independently controllable factors by interacting with the world, (Jain et al., 2021) studied how different actions relate through their effects on the system and (Whitney et al., 2019) proposed a forward prediction objective for simultaneously learning embeddings of states and action sequences. Our work shares the spirit of these approaches but is grounded in a rigorous theoretical framework that provides provable disentanglement guarantees, and is specifically tailored to physical experimental settings rather than RL environments. Looking forward, we see the integration of our framework into RL pipelines, where agents actively select actions or experiments to maximize disentanglement, as a promising direction for future work.

## 2 Theory of action-induced representations

We consider observations of a physical system $x$, assumed to lie in a continuous input data manifold $X \subseteq \mathbb{R}^{d_x}$ and sampled from a probability distribution $p_{\text{data}}$. We further consider that the input manifold $X$ has a much lower dimension than $d_x$, a widely-accepted tenet in computer science referred to as the Manifold Hypothesis (Whiteley et al., 2025). Its coordinates are typically referred to as *factors of variation* $c \in C$ and uniquely describe any given data point $x \in X$. The goal of our proposal is to obtain a disentangled representation of these factors of variation without relying on independent component analysis.

To do so, we take an operational approach and consider a finite set of integers $\mathbb{A} = \{1, 2, \ldots, n_A\}$ which label actions $a$ (also referred to as experiments or tasks in the following). Some of these actions can be implemented together, giving us a subset $P_{\mathbb{A}} \subseteq \mathcal{P}(\mathbb{A})$ specifying allowed actions and action combinations, where $\mathcal{P}(\mathbb{A})$ is the power set of $\mathbb{A}$. For each allowed element $A \in P_{\mathbb{A}}$ and for each input data point $x$, we consider that there is a unique, deterministic ground truth $y_A \in Y_A \subseteq \mathbb{R}^{d_Y}$ (see Figure 1a). Here, $Y_A$ is assumed to be a continuous manifold embedded in a raw output data space $\mathbb{R}^{d_Y}$, similarly as for the input data.

**Example 1.** *Consider an experiment $a = 1$ measuring the gravitational force acting on a resting object, i.e. $y_1 := F_G = mg$ with $g$ the constant gravitational acceleration, which only depends on the mass $m$. An experiment $a = 2$ measures the force exerted by a fixed, homogeneous electric field $E$, given by $y_2 := F_E = qE$ and therefore only depends on the charge $q$. Moreover, elementary actions $a = 1, 2$ can be combined into one experiment $A = \{1, 2\}$, measuring first the gravitational force and then the electrical force. Then, the set of allowed action combinations is $P_{\mathbb{A}} = \{\{1\}, \{2\}, \{1, 2\}\}$. Note that the form of the data $Y_A$ is not relevant as long as it is a continuous manifold. For instance, we could have also used the object's trajectory in the respective fields as the outcomes $y_i$. An implementation of a very similar scenario is presented in Section 4.2.*

This setting gives rise to a collection of datasets as follows:

**Definition 1.** *Let $X \subseteq \mathbb{R}^{d_x}$ be the continuous manifold of the set of input data points, and $\mathbb{A} = \{1, 2, \ldots, n_{\mathbb{A}}\}$ the set of possible elementary actions. Consider a set $P_{\mathbb{A}} \subseteq \mathcal{P}(\mathbb{A})$ specifying the allowed action combinations.*

*For each allowed action combination $A \in P_{\mathbb{A}}$, let there be a deterministic ground truth $y_A : X \to Y_A \subset \mathbb{R}^{d_{Y_A}}, x \mapsto y_A(x)$ such that $Y_A$ is also a continuous manifold.*

*We call $D_A := \{(x, y_A(x)) \mid \forall x \in X\}$ an action-induced dataset for $A \in P_{\mathbb{A}}$.*

The notation for this section is summarized in Figure 1b. From the following, we see that an action-induced dataset has some inherent structure: in Example 1, to reproduce the data $y_1(x)$, we do not need to know the charge $q$, and we do not need to know $m$ to reproduce $y_2(x)$. This motivates the concept of action-induced representations, which may filter unused parameters for a given element of $P_{\mathbb{A}}$:

**Definition 2.** *Let $D_A \subset X \times y_A(X)$ be an action-induced dataset for $A \in P_{\mathbb{A}} \subseteq \mathcal{P}(\mathbb{A})$ where $P_{\mathbb{A}}$ is the set of allowed action combinations and $\mathbb{A} = \{1, 2, ..., n_{\mathbb{A}}\}$ is the set of elementary actions.*

*Consider a continuous manifold $Z \subset \mathbb{R}^{d_Z}$, referred to as a latent space. Let $I_A \subseteq \{1, ..., d_Z\}$ with $A \in P_{\mathbb{A}}$ denote a subset of indices in $\mathbb{R}^{d_Z}$. Given such an index set, we introduce the projector $\pi_A : Z \to Z_A \subseteq \mathbb{R}^{|I_A|}$,*

$$\pi_A(z) \equiv z_A := (z_i)_{i \in I_A} \tag{1}$$

*where $Z_A := \{(z_i)_{i \in I_A} | z \in Z\}$.*

*Let us assume that there exist continuous maps $\psi : X \to Z$ and $\phi_A : Z_A \to Y_A$ for all $A \in P_{\mathbb{A}}$ such that,*

$$(\phi_A \circ \pi_A \circ \psi)(x) = y_A(x) \ \forall x \in X. \tag{2}$$

*Then, we refer to $(Z, \psi, (I_A, \phi_A)_{A \in P_{\mathbb{A}}})$ as an action-induced representation (AIR) of $\{D_A\}_{A \in P_{\mathbb{A}}}$.*

Figure 1b, lower row, illustrates a mapping $\psi(x)$ that transforms an input $x$ into a latent representation $z$ (shown in blue). Given an action combination $A \in P_{\mathbb{A}}$, the corresponding AIR $z_A$ (shown in yellow) is obtained by applying the projection $\pi_A^{(z)}$ to $z$. As we will further elaborate in Section 3, AIRs fit naturally into the framework of VAEs, where $\psi$ represents an encoder, $Z$ the image of the encoder, i.e., its latent space, and $\phi_A$ represents a decoder.

As we have motivated above for the Example 1, AIRs can give rise to an operational form of disentanglement. However, AIRs themselves do not guarantee it, so one needs to further restrict to what we define as *minimal AIR*:

**Definition 3.** *An AIR $(Z, \psi, (I_A, \phi_A)_{A \in P_{\mathbb{A}}})$ is a minimal AIR (minAIR) if the following conditions are satisfied:*

1. *$\psi : X \to Z$ is surjective and $d_Z = |\bigcup_{A \in P_{\mathbb{A}}} I_A|$. All $\phi_A : Z_A \to Y_A$ are invertible and the inverses are continuous.*

2. *$Z$ is open in $\mathbb{R}^{d_Z}$, and uses the standard ($\epsilon$-ball) topology of $\mathbb{R}^{d_Z}$*

In simple terms, condition (1) implies that minAIRs are a faithful parameterization of the $Y_A$. Our requirement that the model can perfectly solve all tasks requires that the decoder can predict all outcomes $y_A$,

and therefore surjectivity. Injectivity is a form of minimality that guarantees that to every $y_A$, there is only one $z_A$, not several. This bijectivity induces identifiability in the sense that the $y_A$ completely specify their latent factors of variation. Therefore, it is a common assumption in the disentanglement literature Rolinek et al. (2019); Khemakhem et al. (2020). Condition (2) imposes that all $z$-entries are free parameters (without redundancies). To be more concrete, Condition (2) requires $(Z)$ to contain a small neighbourhood around each of its points in every latent direction. Consequently, each latent entry can be varied slightly while all other entries are held fixed. This rules out redundant entries constrained by the remaining ones: for example, $(z_2 = z_1)$ restricts a nominally two-dimensional representation to a line, which is not open in $(\mathbb{R}^2)$. This coordinate-wise freedom is used in the proof of Theorem 1 to vary non-overlapping entries independently. Detailed motivation and counter-examples are given in Appendices A and C.

With this definition at hand, we can revisit the Example 1 dataset and discuss some minAIRs:

**Example 2.** *Consider the data from Example 1. An acceptable minAIR would be $z = (m, q)$, with $\phi_1^{(z)}(m) = mg$, $\phi_2^{(z)}(q) = qE$, $\phi_{1,2}^{(z)}(m, q) = (mg, qE)$. Under the assumption that $m > 0$, all of these functions are continuous with continuous inverses. Similarly, $c = (m^2, 2q)$ is a valid minAIR. A non-example is $v = (m, m + q)$, which is not a minAIR because $Y_2$ is one-dimensional, but $Z_2$ is two-dimensional, and therefore $\phi_2$ cannot be invertible. However, if one changes the setup and considers a dataset that only includes $P_{\mathbb{A}} = \{\{1\}, \{1, 2\}\}$, then, $v$ becomes a minAIR with $\phi_1^{(v)} = \phi_1^{(z)}$ and $\phi_{1,2}^{(v)}(m, m + q) = \phi_{1,2}^{(z)} \circ \varphi(m, m + q)$ where $\varphi(m, m + q) = (m, q)$ is invertible*

**Operational disentanglement in minAIRs** We now show that minAIRs have a very specific structure that explicitly *disentangles* parameters w.r.t. their action-dependence. This disentanglement is summarized by our main theorem:

**Theorem 1.** *Consider two minAIRs, $\left(Z, \psi^{(z)}, (I_A^{(z)}, \phi_A^{(z)})_{A \in P_{\mathbb{A}}}\right)$ with convex set $Z$ and $\left(C, \psi^{(c)}, (I_A^{(c)}, \phi_A^{(c)})_{A \in P_{\mathbb{A}}}\right)$ with convex set $C$. For any nonempty set of action combinations $\mathcal{A} := \{A_1, ..., A_m\} \subseteq P_{\mathbb{A}}$, let $I_{\mathcal{A}}^{(z)} = \bigcap_{A \in \mathcal{A}} I_A^{(z)}$ (and analogously $I_{\mathcal{A}}^{(c)}$) be the index sets of latent vector components they have in common.*

*Then, for any $A_j \in \mathcal{A}$, the map $g_{\mathcal{A}} : Z_{A_j} \to C_{\mathcal{A}}$,*

$$g_{\mathcal{A}} := \pi_{\mathcal{A}}^{(c)} \circ (\phi_{A_j}^{(c)})^{-1} \circ \phi_{A_j}^{(z)} \tag{3}$$

*only depends on the overlap $z_{\mathcal{A}}$, but not on $j$ and not on the completion to $z_{A_j}$. It is bijective as a function of $z_{\mathcal{A}} \in Z_{\mathcal{A}}$ only.*

The proof of the Theorem and a more general version that does not assume full convexity of $Z$ and $C$ is given in Appendix B. This Theorem implies that the neurons that are shared for different action combinations (those in $I_{\mathcal{A}}$) will be disentangled from the rest for any minAIR (see Figure 1c). In other words, it shows that the $z_{\mathcal{A}} \in Z_{\mathcal{A}}$ and $c_{\mathcal{A}} \in C_{\mathcal{A}}$ uniquely determine each other via $g_{\mathcal{A}}$ through their mutual connection to $y_A \forall A \in P_{\mathbb{A}}$ (see Figure 1b). In this sense, disentanglement is not directly related to the cardinality of $P_{\mathbb{A}}$, but rather to whether the actions produce intersections $I_{\mathcal{A}}$ that isolate individual factors $c_i$. Importantly, such disentanglement is met only when our considered assumptions hold (see Appendix A and Appendix D).

**Example 3.** *Consider the dataset from Example 1 and the minAIRs in Example 2. Notably, for $P_{\mathbb{A}} = \{\{1\}, \{2\}, \{1, 2\}\}$ all minAIRs disentangle $m$ and $q$ (Figure 1c, middle scheme), while for $P_{\mathbb{A}} = \{\{1\}, \{1, 2\}\}$, they only disentangle $m$, since $v$ is also a minAIR (Figure 1c, bottom scheme). The reduced disentanglement in the latter case is exactly a consequence of $m$ being shared by both action combinations $A_1 = \{1\}$ and $A_{1,2} = \{1, 2\}$ while $q$ alone is in no intersection.*

## 3   VAIR: a model for minAIR

Given the problem stated above, we introduce an architecture, called variational AIR (VAIR), designed to approximate such minAIRs and hence, for suitable sets of actions, disentangle the hidden factors of variation of the input data.

**Background on VAE.** The proposed model is based on the variational autoencoder (VAE) (Kingma & Welling, 2013), which consists of an encoder $q_\phi$ that compresses input samples $x$ into a latent representation $z$, and a decoder $p_\theta$ that reconstructs the original input from this latent space. In VAEs, latent variables are probabilistic neurons whose distribution is typically modeled by a multivariate Gaussian. In practice, the encoder outputs the mean $\mu_i$ and variance $\sigma_i^2$ of the latent variables from which the values for each latent neuron $z_i$ are sampled and then passed to the decoder. The model is trained via the Evidence Lower Bound (ELBO),

$$\mathcal{L} = \frac{1}{N} \sum_{n=1}^{N} \mathbb{E}_q[\log p_\theta(x^{(n)} \mid z)] - \beta \mathrm{KL}[q_\phi(z \mid x^{(n)}) || p(z)] \tag{4}$$

with $N$ being the number of samples $x^{(n)}$ in the training set and $\beta$ the regularization parameter introduced in  (Higgins et al., 2017). The loss function introduced in Equation (4) contains two terms. The first is a reconstruction loss, which enforces similarity between the predicted and target outputs. In standard VAEs, this corresponds to the likelihood $p_\theta(x^{(n)} \mid z)$ of reconstructing the input $x^{(n)}$ from the latent representation. In this work, the same formulation is used, but the target output is typically the experimental outcome $y^{(n)}$. While VAEs are typically trained to reconstruct their input, modern variants have also considered settings where the output differs from the input (Sohn et al., 2015; Iten et al., 2020; Nautrup et al., 2022), naturally falling into the Information Bottleneck principle (Tishby & Zaslavsky, 2015). The second term is a regularization loss, given by the Kullback-Leibler divergence between the approximate posterior $q_\phi(z_i \mid x^{(n)}) \sim \mathcal{N}(\mu_i, \sigma_i)$ and a standard normal prior $p(z) = \mathcal{N}(0, 1)$. This term encourages the latent representation to align with the prior, while competing with the reconstruction objective.

This trade-off results in a polarized latent space (Rolinek et al., 2019), where only the minimal set of neurons required for reconstruction deviates from the prior (*active* neurons), while the remaining ones collapse to it (*passive* neurons). Additionally, an intermediate regime of *mixed* neurons, active only for subsets of the data, has been identified (Bonheme & Grzes, 2023).

**VAIR: adapting VAE for AIR.** VAIR is a variant of VAE purposely built to approximate minAIR and hence produce disentangled representations by Theorem 1. As VAE, VAIR is trained following the ELBO in Equation (4). The main change compared to standard VAE is the use of two separate encoders (see Figure 1d). The first, $E_X$, takes the observations $x$ and outputs only the mean $\mu_i$ of the latent neurons. The second, $E_A$, takes the action combination $A \in P_\mathbb{A}$ and outputs their variances $\sigma_i^2$, which determine which neurons are noised out for a given action. For brevity, we often use elementary actions $a \in \mathbb{A}$ and action combinations $A \subseteq P_\mathbb{A}$ interchangeably. Following the common definitions of the polarized regime, we call neurons active if $\sigma_i^2 \to 0$ for all actions. As defined in Definition 2, for some AIRs not all latent neurons are required for every action $a$, and the regularization in Equation (4) pushes the model to noise out the unnecessary ones. Neurons that are active only for a subset of actions are referred to as *mixed* neurons.

Finally, we feed the latent vector to the decoder $D$. Moreover, because the action cannot be inferred from the sampled latent vector, and is indeed needed to reconstruct $y_a$, we also input it to $D$. We note here that compared to the latent model of Equation (4), VAIR defines a likelihood $p_\theta(y^{(n)}|z, a^{(n)})$ and a latent posterior $q_\phi(z|x^{(n)}, a^{(n)})$, which can be nonetheless optimized in the same fashion. Interestingly, related conditioning ideas appear in iVAE (Khemakhem et al., 2020). There, identifiability relies on a context-dependent prior $p(z|a)$, while our approach instead uses action-dependent encoder and decoder components to induce disentanglement, without modifying the prior.

Given the previous discussion, we identify $E_X(x)$ with the mapping $\psi$ (Section 2), which transforms the input $x$ into a compressed latent representation $z$. Similarly, the encoder $E_A(a)$ corresponds to $\pi_{\{a\}}$, adapting the latent space to the specific input action. Finally, the decoder $D(z, a)$ plays the role of $\phi_{A=\{a\}}$, transforming the resulting latent vector into the output associated with that action. While this comparison is heuristic, we provide numerical evidence for the emergence of minAIR in VAIR in Section 4. We note that, while Section 2 considered discrete actions, numerical experiments presented below demonstrate that minAIRs can also be achieved for continuous actions (see Section 4.3).

Moreover, we note that the quality of disentanglement induced by VAIR is contingent on the precision of the action representation fed into $E_A$; noisy or partially observed actions will result in uncertain variance

estimates which could prevent disentanglement by AIR, although we will show robustness to noise and generalization capabilities in Section 4.2 and Section 4.3.

**Why does VAIR approximate minAIRs?** We now give a heuristic interpretation of why VAIR is expected to approximate minAIRs. In the polarized regime, latent neurons can be idealized as either noise-free or noised-out, depending on the action combination $A \in P_{\mathcal{A}}$. The noise-free coordinates then play the role of the index set $I_A$, while the decoder restricted to these coordinates approximates $\phi_A$. Since the KL regularization favors noising out coordinates that are unnecessary for predicting $y_A$, VAIR encourages small action-specific latent spaces and, after globally passive neurons are discarded,

$$d_Z = \left| \bigcup_{A \in P_{\mathcal{A}}} I_A \right|.$$

These pressures do not, however, guarantee all minAIR assumptions. Information may remain duplicated across coordinates or different active coordinates may be functionally dependent, in which case $Z$ is not open in its embedding space. Moreover, deterministic and accurate decoding does not by itself imply that $\phi_A$ is invertible; this additionally requires the task outputs to uniquely identify the relevant factors of variation. Thus, VAIR is designed to approach the minAIR conditions, but Theorem 1 applies only to the extent that these conditions are satisfied. Illustrative counter-examples for cases where not all assumptions are satisfied are given in Appendix C. Revealing the underlying factors therefore requires a sufficiently informative and carefully designed collection of experiments: no learning procedure can identify factors that are not distinguished by the available experiments.

## 4 Numerical experiments

In the following, we will train VAIR on different datasets that will showcase different aspects, strengths and potential weaknesses. Section 4.1 introduces an abstract experiment, designed to validate the theoretical predictions of Theorem 1 and compare VAIR against state-of-the-art VAE disentanglement approaches. We then consider two scientifically motivated settings: first, Section 4.2 presents a classical physics experiment demonstrating how VAIR reveals physically meaningful representations, while also studying the effect of uncontrollable external perturbations on the learned latent structure; second, Section 4.3 tests VAIR in a real-world quantum physics problem, where we further explore the ability of VAIR to generalize to unseen observations and actions. Code for reproducing the results of this section can be found at https://github.com/anom-tmlr-air/air.

### 4.1 Abstract experiment

We start by considering an abstract problem that will allow us to illustrate the various theoretical concepts presented in Section 2. For that purpose, we define a simple dataset arising from four factors of variation $c_1, \ldots, c_4$. An observation $x \in \mathbb{R}^{d_x}$ is then given by some fixed random functions $F$ (see Appendix E.1). The outputs $y \in \mathbb{R}^{d_y}$ are also calculated from fixed random functions $G, G'$ as $y_{a_1} = G(c_1, c_2)$ and $y_{a_2} = G'(c_2, c_3, c_4)$ excluding action combinations. Hence, $\dim Z = 4$, $\dim Z_{a_1} = 2$ and $\dim Z_{a_2} = 3$ (see Figure 2a).

We now set $d_x = d_y = 10$ and train a VAIR with 6 latent neurons (see Appendix F). In Figure 2 we show the evolution of the latent neurons' variance predicted by $E_A$ for each of the two input actions. Already in the early stages of training, the model learns that only four neurons are required to reconstruct the output, setting $\log \sigma_2^2, \log \sigma_4^2 \to 0$ for both actions, hence making them passive neurons. Then, the model encodes information needed for action $a_1$ in two neurons, namely 1 and 3, as shown by the variance decrease in the top panel, and in three neurons for action $a_2$. That is, we can see that the model retains four non-passive neurons, from which three (1,5,6) are mixed, as they are only active for either action 1 or 2, and only one is active (3). This is in accordance with expected results for $\dim Z = 4$, $\dim Z_{I_{a_1}} = 2$ and $\dim Z_{I_{a_2}} = 3$.

The fact that neuron 3 is an active neuron comes from it being in the intersection $I_{a_1} \bigcap I_{a_2}$. By Theorem 1, and given that $c_2$ is at the intersection of the underlying experiments (Figure 2a), any minAIR must disentangle such factor of variation. To show this, we plot in Figure 2c the latent neurons' means as a function

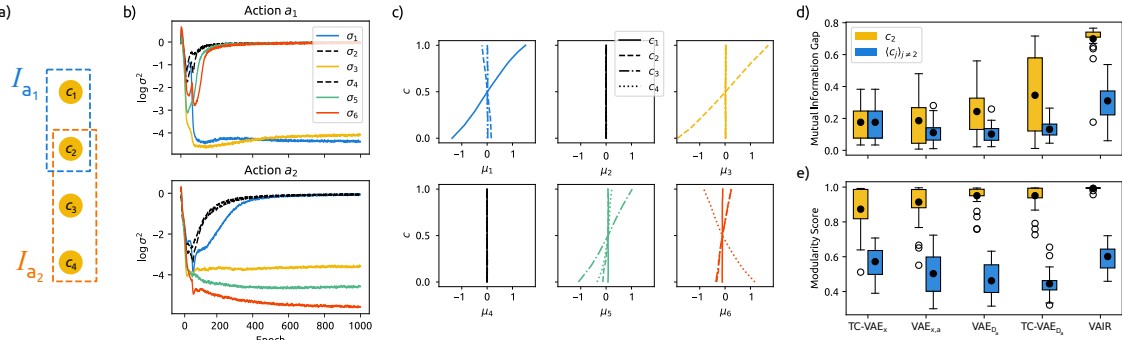

Figure 2: **Abstract experiment a)** Factors of variation in the experiment, indicating the subsets $I_a$ associated with each of the considered actions. **b)** Latent neuron variances $\sigma_i$ output by $E_a$ for the two actions in the experiment, plotted as a function of training epoch. **c)** Latent neuron means $\mu_i$ output by $E_x$ as a function of the hidden factors $c_i$. Colors correspond to the neurons (as in panel b), while line styles indicate different factors. **d, e)** Mutual information gap (MIG) and modularity score, respectively, computed for two groups of factors: $c_2$, and all remaining factors excluding $c_2$. Boxplots show results from 20 independent training runs per model, each with random dataset generation and weight initialization.

of the different factors' values. For each line, we fix all $c_{j\neq i}$, and change the value of $c_i$ monotonically in its range. As expected, due to $z_2$ and $z_4$ being passive, both means are constant at zero. Conversely, the rest show a clear dependence w.r.t. different $c_i$. In particular, $\mu_3$ is only dependent on $c_2$, demonstrating that this neuron encodes a disentangled representation. On the other hand, $\mu_1$ encodes mainly a representation of $c_1$, but also has a slight dependence on $c_2$. We recall that VAIR is not constructed to create a disentangled representation for factors outside any intersection, and that any other disentanglement mostly depends on the training and the inherent bias of VAEs to disentangle (Bhowal et al., 2024). Indeed, we see that neurons 5 and 6 encode an entangled representation of the factors needed to reconstruct $y_{a_2}$.

A higher-dimensional version of the current example is presented in Appendix H, showcasing the ability of VAIR to reach disentangled representations in more complex scenarios.

**Comparison with other VAE variants** We compare VAIR to four VAE variants: (i) TC-VAE$_x$, a $\beta$VAE trained without actions using the Total Correlation (TC) loss (Chen et al., 2018); (ii) VAE$_{x,a}$, a $\beta$VAE receiving both state and action as input; (iii) VAE$_{D_a}$, which also feeds the action to the decoder, partially mirroring VAIR's structure; and (iv) TC-VAE$_{D_a}$, identical to the previous but trained with the TC-loss. While VAE$_{x,a}$ must encode the action in the latent space, violating the AIR assumptions, VAE$_{D_a}$ and TC-VAE$_{D_a}$ do not, and could in principle approximate minAIRs. All models reach similar reconstruction losses, confirming that the observed differences in disentanglement shown below are not due to tradeoffs with reconstruction quality, but rather reflect genuine differences in the learned latent representations. See Appendix G for details.

We first evaluate the learned representations using the mutual information gap (MIG) between the factors of variation $c_i$ and the mean of each latent neuron $\mu_k$ (Chen et al., 2018), with larger MIG values indicating more disentangled representations. In Figure 2d, we plot the MIG scores for two subsets of neurons. The blue bars show the MIG corresponding to $c_2$, the factor expected to be disentangled from the experiment's construction. For TC-VAE$_x$, as no actions are considered, we average over all $c$, showcasing the inability of VAE to fully disentangle the dataset (Locatello et al., 2019). On the other hand, as predicted by Theorem 1 and observed in Figure 2c, VAIR achieves near-complete disentanglement of this factor. VAE$_{D_a}$ and TC-VAE$_{D_a}$ also show improved disentanglement of $c_2$ compared to the standard VAE, although they still perform significantly worse than VAIR. This is due to their inability to dissociate the action from their encoded representation, which also further hinders the MIG when averaged over the rest of the variables $\langle c_j \rangle_{j\neq 2}$ (see Appendix G.2 for further details on this). Similarly, the scores drop substantially even for VAIR, as it is not the objective of the architecture to promote disentanglement of components beyond the intersection structure of the action set. For broader disentanglement, VAIR can be combined with existing VAE disentanglement approaches;

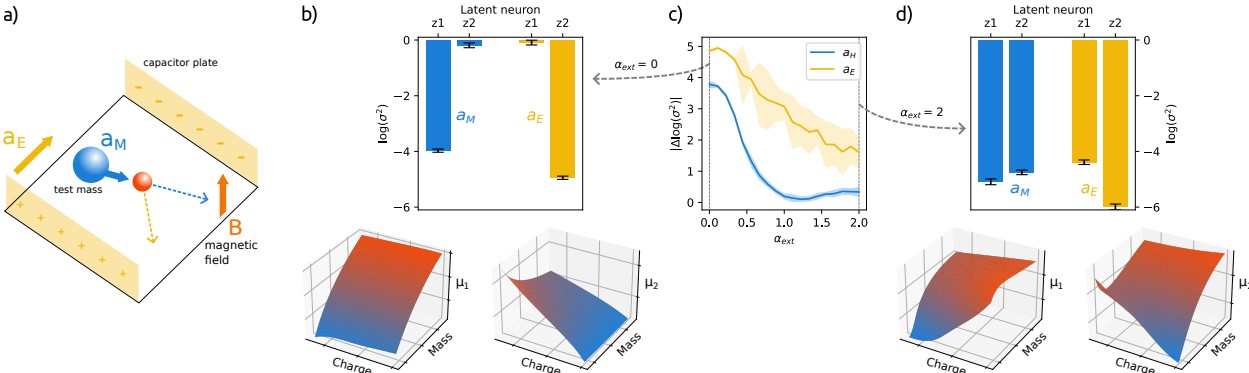

Figure 3: **Classical experiment a)** Schematic representation of the experiment: we consider objects of certain mass and charge and two actions: an elastic collision with a larger object with fixed mass ($a_M$) *or* the activation of an electric field via a capacitor ($a_E$). We also consider the additional presence of an external magnetic field $B$ of different coupling strengths $\alpha_{\text{ext}} > 0$. **b)** The central panel shows the learned variances by $E_A$ for the two different actions at $\alpha_{\text{ext}} = 0$, averaged over 20 different models. The two lower plots showcase, for a single model, the relationship between each latent neuron's $\mu_i$, as a function of the input trajectories' mass and charge. **c)** Difference between the predicted variances by $E_A$ for each action, as a function of the strength $\alpha_{\text{ext}}$ of the external magnetic field $B$. **d)** Same as panel b but for $\alpha_{\text{ext}} = 2$.

however, we caution that the additional objectives of such methods may conflict in particular applications with the natural AIRs that emerge from the action structure (see Appendix G.2 and Appendix H).

To complement the previous analysis, we compute the modularity score (Carbonneau et al., 2022). While the MIG measures whether a single generative factor is captured by multiple latent neurons, the modularity score measures whether a single latent neuron captures multiple generative factors. As shown in Figure 2e, the same trend as in the MIG is found: VAIR clearly outperforms other methods when the modularity score is computed for $c_2$, while the results are comparable for the rest of the factors. These results further highlight that VAIR correctly reaches the disentanglement predicted by Theorem 1.

## 4.2 Classical physics experiment

Next, we present a realistic experiment based on classical physics and closely related to the running example in Section 2. We will show the existence of non-trivial minAIRs that differ from our typical physical interpretation of a system and the effect of confounding factors, in the form here of uncontrolled external sources hindering disentanglement by AIR.

We consider an object of mass $m$ and charge $q$ (see Figure 3a). The experiment consists of two actions: 1) the collision action, referred to as *mass* action $a_M$, considers the elastic collision of the object with a fixed, chargeless test mass $M > m$ with velocity $v_H$; 2) the *electric* action $a_E$ consists of the placement of a parallel plate capacitor that generates an electric field $E$. Instead of action combinations, we will also later consider the presence of an additional noise source in the form of a magnetic field $B$ with varying strengths (see in Appendix E.2). The goal of this experiment, and hence the output of VAIR, is to predict the two-dimensional trajectory $y_{a_i}(t)$ as the result for a given action. From this setting, one can see that for $y_H(t)$ only the mass is relevant while for $y_E(t)$ one needs both the mass and charge.

For simplicity, we consider the observation to consist of the stacked trajectories resulting from the actions, $x = \{y_H(t), y_E(t)\}$. An alternative choice would have been, for example, a previous observation of the object's dynamics. Crucially, in line with the problem formulation in Section 2, all factors of variation must be recoverable from such observations. If one were to train on a single trajectory at a time instead, the encoder would be able to infer which action was applied. As a result, VAIR may construct representations that differ from minAIRs.

We now train on such a dataset VAIRs with two latent neurons, for simplicity and due to the expected $\dim Z = 2$. As shown by the final learned variances presented in Figure 3b, averaged over 20 different models, the two neurons are mixed, each remaining active for one action but not the other. While this was expected for the mass action, as a single factor $(m)$ is needed for $y_H(t)$, we expected both $m$ and $q$ to be required to predict $y_E(t)$. By inspecting the learned representations (side panels in Figure 3b), we see that while $\mu_1$ encodes $m$ and is independent of $q$, $\mu_2$ encodes a function of both $m$ and $q$. In particular, it encodes a representation proportional to $q/m$, the factor defining the trajectory of a massive, charged object in an electric field (see Appendix E.2). Indeed, the learned representation $z = (m, q/m)$ is a minAIR, and the expected $c = (m, q)$ is not, as $\dim Z_{I_{a_E}^{(z)}} = 1 < \dim Z_{I_{a_E}^{(c)}} = 2$. This highlights the importance of actions and their effect on a physical system for our physical understanding. In the current experiment, due to the nature of the set of actions, it is not possible to isolate $q$, and hence one must search for a better experimental setup that would fulfill this goal. We further discuss this in the conclusion.

**External forces** In many experimental scenarios, the presence of confounding factors, in the form here of external, uncontrollable perturbations, hinders the correct interpretation of the observations. To illustrate this within the AIR framework, we consider the presence of an external magnetic field with relative strength $\alpha_{\text{ext}}$ (see Appendix E.2) and train models at different strengths. Importantly, such an external force introduces the necessity of encoding both factors $m$ and $q$ for both actions. In Figure 3c, we show the difference of the variances for each action. Here, a high variance means that one neuron is more important for one action than for the other. Focusing on $a_M$ (blue line), we observe that at $\alpha_{\text{ext}} = 0$ the difference is maximal, due to a single neuron sufficing to describe the output of this action. For larger $\alpha_{\text{ext}}$, the output of $a_M$ depends on both the mass and the charge. Consequently, a second neuron becomes active, leading to a decrease in $\sigma_2^2$ and, therefore, a smaller variance difference.

Interestingly, the decrease in $\sigma^2$ does not occur abruptly, since the effect of the field is negligible for small $\alpha_{\text{ext}}$ and barely influences the output trajectories. As its contribution increases, the latent space must begin to encode this variable, as neglecting it would prevent accurate reconstruction and lead to a higher reconstruction loss. This results in a gradual decrease of the variance difference, reflecting the growing influence of the field on the output trajectories. The same trend is observed for $a_E$. For larger $\alpha_{\text{ext}}$, both latent neurons are active, as shown in Figure 3d. The side panels show that now both neurons encode a mixed representation of the factors of variation. This highlights the effect of uncontrollable factors, such as the magnetic field here, on the learned representations. Indeed, due to the external force both $c = (m, q/m)$ and $z(m, q) = (z_1(m, q), z_2(m, q))$, with $z$ being an invertible function, are minAIRs, and hence the model has no preference between them.

### 4.3 Quantum physics experiment

We now consider a quantum physics experiment related to quantum tomography (Paris & Rehacek, 2004). The latter aims to characterize a quantum state by means of a set of measurements. This example will help us showcase the generalization capabilities of the action encoder $E_A$ to unseen actions and to action combinations. In particular, we will show that $E_A$ is capable to adapt the noise levels in the latent space when receiving test set actions, and that the $\sigma^2$ values predicted directly relate to the identified properties of the actions.

Here we consider a dataset of 2-qubit quantum states, each uniquely described by a certain density matrix $\rho$, a complex, Hermitian $4 \times 4$ matrix with trace 1. We then select 75 random projective measurements $\{|\psi_j\rangle \langle \psi_j|\}_{j=1}^{75}$, fixed before training. From these, each observation $x$ is the collection of probabilities $p_j$ of finding the system described by the density matrix $\rho$ in each of the states $|\psi_j\rangle$, namely $x = \{p_j = \langle\psi_j| \rho |\psi_j\rangle \mid p_j \in [0,1]\}_{j=1}^{75}$. The task consists of predicting the expectation value of a selected Pauli projector $a_{i,k} = w_{i,k}$, $y_{a_{i,k}} = \text{tr}(w_{i,k}\rho)$, where $w_{i,k}$ is defined below.

From this setting, we can deduce that the dataset has 15 real-valued factors of variation, as any state $\rho$ can be described by a complex $4 \times 4$ matrix, while one element of the matrix can be calculated from the normalization constraint. To facilitate the interpretation of the latent space, we define $\{w_{i,k}\}_{i,k=0}^{3} \setminus \{w_{0,0}\}$ as the set of projectors onto the $+1$ eigenstates of the two-qubit *Pauli* operators, i.e. $w_{i,k} = (\mathbb{I} + \hat{\sigma}_i^{\text{I}} \otimes \hat{\sigma}_k^{\text{II}})/2$,

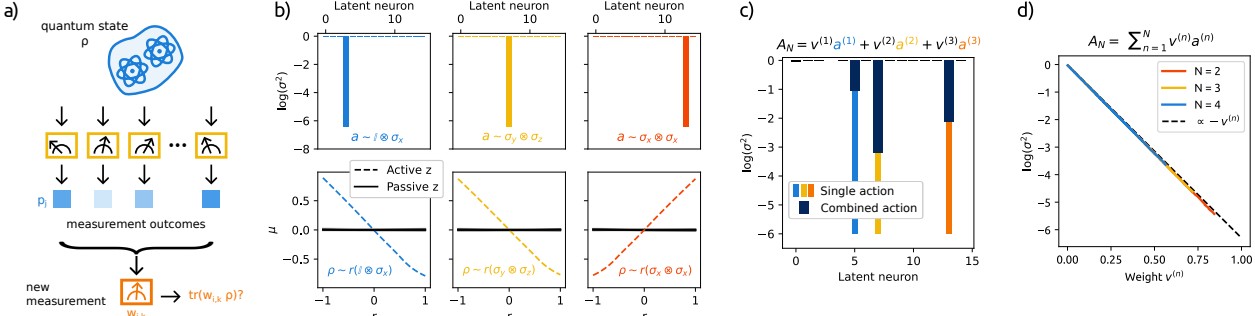

Figure 4: **Quantum experiment a)** Schematic representation of the experiment: a quantum state $\rho$ is repeatedly measured. From the measurement outcomes, the goal is to predict the outcome of a new, given measurement of $\rho$. **b)** Upper plots: output variances of $E_A$ for different actions in the training set, showing that a single neuron is active for each action. Lower plots: output means of $E_X$ for the same active neurons (dashed) as a function of the tuning parameter $r$. Passive neurons (solid) are also plotted, and remain constant for all values of $r$. **c)** Output variances of $E_A$ for an action combination $A_N$ with $N = 3$ (dark blue). Other colors show the output when each individual action $a^{(n)}$ from the combination is independently fed into $E_A$. **d)** Output variances of $E_A$ for action combinations $A_N$, as a function of the parameter $v^{(n)}$, restricted to neurons active for the respective $a^{(n)}$. Dashed line shows the scaling $\log(\sigma^2) \propto -v^{(n)}$. Results are averaged over 14 different models. Error bars are too small to be visible.

where $i, k = 0, ..., 3$ denote the different Pauli matrices and superindices I, II denote the qubit indices 1 and 2. These projectors $w_{i,k}$ are the inputs to $E_A$. More details can be found in Appendix E.3.

Training VAIR on such a dataset we obtain 15 mixed neurons. As shown in Figure 4b, upper panels, each neuron is active only for a single action, while the others remain at $\log \sigma^2 \to 0$. To explore the learned representations, we create a test set of states of the form $\rho(r) = \frac{1}{4} \left( \mathbb{I} + r(\hat{\sigma}_i^{\mathrm{I}} \otimes \hat{\sigma}_k^{\mathrm{II}}) \right)$. As shown in the lower panels of Figure 4b, each $\mu_i$ responds solely to a given $\hat{\sigma}_i \otimes \hat{\sigma}_k$, meaning that the VAIR has exactly learned the Bloch representation of the qubit state (Gamel, 2016). We again note that this is an action-induced representation in accordance with minAIR, and that choosing a different set of actions, as e.g., a different combination of Pauli matrices, will result in a similar pattern of mixed neurons, but a completely different representation in the latent mean values.

**Combinations of actions**  We now explore the generalization capabilities of the action encoder $E_A$ to accurately represent a given combination of actions. To this end, we consider actions of the type $A_N = \sum_{n=1}^{N} v^{(n)} a^{(n)}$ with $v^{(n)} \in [0, 1]$ and $\sum_{n=1}^{N} v^{(n)} = 1$ and randomly chosen $a^{(n)} = a_{i,k}$. In Figure 4c we show the variances predicted for a given $A_N$ with $N = 3$ (dark blue). We also show the variance for each separate action $a^{(n)}$ composing $A_N$. As shown, the encoder conveys the additive nature of the action and activates the necessary latent neurons even though it has not been trained on action combinations. Note that the combined action $A_N$ is not a physical measurement and cannot be used to calculate output probabilities, i.e., $y_{A_N}$ is undefined. Nevertheless, it is useful for the purpose of interpreting the results in the latent space.

Interestingly, the value of the predicted variance directly depends on the weight of a particular action. To show this, we consider different $A_N$ with $N = 2, 3, 4$ and track the predicted variance for the neuron encoding the corresponding $a^{(n)} = a_{i,k}$ w.r.t. the value $v^{(n)}$, showcasing a surprisingly consistent linear behavior (see Figure 4d, where we consider the average over 14 distinct models). In particular, as $v^{(n)}$ increases, $\sigma^2$ decreases, as the decoder needs further information from the neuron encoding the corresponding $a^{(n)}$ to correctly reconstruct its output. This behavior shows that the encoder $E_A$ balances the noise of the latent neuron according to the action weight on the output, and most importantly that it generalizes beyond the training set of single actions to action combinations.

## Conclusion

In this work, we introduced *action-induced representations* (AIR), a framework for recovering disentangled representations of physical systems through their dependence on actions describing experimental settings. We provided theoretical results showing that minimal AIRs (minAIRs) provably disentangle the system's degrees of freedom based on their action-dependence. This disentanglement arises from the overlap between actions and the latent variables they influence, guaranteed by Theorem 1.

To realize this framework in practice, we proposed *VAIR*, a VAE-based architecture specifically designed to approximate minAIRs. The model leverages a dual-encoder structure in which one encoder extracts latent representations of the system state, while the other controls which latent variables are active for a given action. Across a range of experiments, from abstract benchmarks to classical and quantum physics simulations, we demonstrated that VAIR consistently outperforms standard VAE architectures in recovering disentangled and interpretable latent variables. Moreover, we showed that the architecture allows us not only to extract the system's hidden properties, but also to interpret the role of actions, by identifying the action-dependence of each degree of freedom.

Looking forward, we see several promising directions. One key avenue is to integrate VAIR into reinforcement-ment learning pipelines, where agents could actively select the actions that maximize disentanglement or information gain, effectively guiding experimentation towards interpretable representations. Additionally, while our current analysis focuses on discrete or fixed action sets, extending the theoretical guarantees to continuous or high-dimensional action spaces remains an open and relevant challenge. In a similar direction, joint-embedding predictive architectures (JEPAs) (LeCun et al., 2022) align naturally with our state-action framework presented here, and its combination with AIR is a promising direction for scaling the latter guarantees to high-dimensional problems. Finally, we believe that embedding architectures like VAIR into real-world experimental systems could aid the autonomous discovery of novel phenomena in physical systems by guiding researchers not only to extract minimal representations of their data, but also to understand the effects different experimental interventions have over the system.

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

# A Motivation of the axioms for minAIRs

In this appendix section, we motivate our assumptions for minAIRs. They express what an ideal minimal parameterization of the task manifolds $Y_A$ should be like. For readability, we state the definition here again:

**Definition 4.** *An AIR $(Z, \psi, (I_A, \phi_A)_{A \in P_\mathbb{A}})$ is an* ideal *or* minimal AIR *(minAIR) if the following extra conditions are satisfied:*

1. $\psi : X \to Z$ *is surjective.*

2. $d_Z = |\bigcup_{A \in P_\mathbb{A}} I_A|$.

3. *All $\phi_A : Z_A \to Y_A$ are invertible and the inverses are continuous.*

4. *$Z$ is open in $\mathbb{R}^{d_Z}$, with respect to the standard ($\epsilon$-ball) topology of $\mathbb{R}^{d_Z}$*

Assumption 1: We demand that $\psi$ is surjective, because the latent space $Z$ should be the image of the encoder, which is $\psi$.

Assumption 2: We require $d_Z = |\bigcup_{A \in P_\mathbb{A}} I_A|$ such that every entry in the parameterization is relevant for some task. Otherwise, we might have some stray factors of variation that do not matter for any of the tasks. Here, minimality is expressed by demanding that we have no factor of variation in our parameterization which is irrelevant for all tasks.

Assumption 3: We require the $\phi_A$ to be surjective such that they can reach every ground truth $y_A$. In particular, surjectivity is required for the minAIR setup to perfectly solve the tasks. Furthermore, they should be injective, or faithful, such that every $y_A$ is uniquely identified by a single $z_A$. In particular, $z_A \neq \tilde{z}_A$ are guaranteed to belong to different $y_A$. Here, minimality is expressed by only having one $z_A$ per ground truth $y_A$, not several. The direct consequence of this notion of minimality is identifiability: If two different latent $z_A \neq z'_A$ give us the same answers, then our model cannot be guaranteed to recover the true values of the underlying factors of variation. Therefore, injectivity is a common assumption in the disentanglement literature (see e.g., Rolinek et al. (2019); Khemakhem et al. (2020)).

Assumption 4: We require $Z \subset \mathbb{R}^{d_Z}$ to be open such that every entry $z_i$ is a free parameter which we can vary without perturbing the other entries. This is the case because every open set in $\mathbb{R}^{d_Z}$ contains an open $\epsilon$-ball around every point in the open set. Here, minimality is expressed as requiring that no $z_i$ is a function of the other entries, and therefore redundant.

To be more specific, if $Z$ is open in its embedding space $\mathbb{R}^{d_Z}$, for every latent point $z^{(0)} \in Z$ there is an open $\epsilon$-ball in $\mathbb{R}^{d_Z}$ that is fully contained in $Z$. Now, within the ball, we construct a point $z$ by first specifying its components $z_1, z_2, \ldots, z_{d_Z-1}$. Since the point is supposed to be in the open ball, its $\ell_2$ norm is smaller than $\epsilon$, i.e. there is a $\delta r > 0$ such that $\sum_{j=1}^{d_Z-1}(z_j - z_j^{(0)})^2 = \epsilon^2 - \delta r^2$. Now, let us consider the remaining component $z_{d_Z}$. The only condition for $z$ to be in the open $\epsilon$-ball is that $\sum_{j=1}^{d_Z}(z_j - z_j^{(0)})^2 < \epsilon^2$. Therefore, every choice of $z_{d_Z}$ satisfying $(z_{d_Z} - z_{d_Z}^{(0)})^2 < \delta r^2$ is fine. In particular, since we can still choose $z_{d_Z}$ from a continuum of values despite the fact that we already fixed all the other $z_j$, this shows that $z_{d_Z}$ is a free parameter on $Z$. Even after fixing all the other parameters $z_1, \ldots z_{d_Z-1}$ they do not determine $z_{d_Z}$.

As a simple example for what can go wrong without openness, consider $d_Z = 2$ and $Z = \{z \in \mathbb{R}^2 | z_1 = z_2\}$. $Z$ is a straight line in 2D, and therefore not open. For any $z \in Z$, if we specify $z_1$, we automatically know $z_2$. Therefore, the coordinate $z_2$ is not a free parameter. A minimal embedding space for a proper minAIR would have been the choice $d_Z = 1$, with $Z = \mathbb{R}^1$.

When considering alternative definitions of minAIRs, one needs to be aware of some pitfalls which we exemplify in Appendix C.

# B  Proof of Theorem 1

In this part of the appendix, we provide a proof of the disentanglement theorem. More specifically, we prove a more general theorem. If $Z$ itself is not convex, it can still be applied to any open, convex subset of $Z$. Since our VAIRs contain a part in the loss function which pushes the latent distribution to be close to the rotationally symmetric normal distribution, we expect that many of the latent spaces encountered in practice will be close to convex.

**Theorem 2** (Action-induced disentanglement, formal and detailed). *Let* $\left(Z, \psi^{(z)}, (I_{A'}^{(z)}, \phi_{A'}^{(z)})_{A' \in P_\mathbb{A}}\right)$ *and* $\left(C, \psi^{(c)}, (I_{A'}^{(c)}, \phi_{A'}^{(c)})_{A' \in P_\mathbb{A}}\right)$ *be minAIRs, and consider a subset* $Z|_{\mathrm{patch}}$ *of* $Z$. *This subset is required to be convex and open in the embedding space* $\mathbb{R}^{d_Z}$.

*Further, let* $\mathcal{A} \subseteq P_\mathbb{A}$ *be a set of allowed action combinations, and define* $I_O^{(z)} := \bigcap_{A' \in \mathcal{A}} I_{A'}^{(z)}$, *as well as* $z_O := \pi_O^{(z)}(z) := (z_j)_{j \in I_O^{(z)}}$ *and* $Z_O := \{\pi_O^{(z)}(z) \, \forall z \in Z\}$. *Analogously for* $c$.[1]

*Then:*
*The function* $g_O : Z_O|_{\mathrm{patch}} \to C_O|_{\mathrm{patch}}$, *with* $Z_O|_{\mathrm{patch}} := \pi_O^{(z)}(Z|_{\mathrm{patch}})$ *and* $C_O|_{\mathrm{patch}}$ *being the image of* $g_O$, *defined as*

$$g_O = \pi_O^{(c)} \circ (\phi_A^{(c)})^{-1} \circ \phi_A^{(z)} \circ i_A, \tag{5}$$

*is well-defined and independent of the choice of* $A \in \mathcal{A}$ *and embedding* $i_A(z_O) = z_A$ *into* $Z_A|_{\mathrm{patch}} := \pi_A^{(z)}(Z|_{\mathrm{patch}})$ *such that* $\pi_O^{(z)}(z_A) = z_O$. *Furthermore,* $g_O$ *is surjective and continuous, and* $C_O|_{\mathrm{patch}}$ *is path-connected.*

*If additionally,* $C|_{\mathrm{patch}} \subseteq C_{\mathrm{conv}} \subseteq C$ *where* $C_{\mathrm{conv}}$ *is a convex and open subset of* $\mathbb{R}^{d_C}$, *then,* $g_O$ *is also a homeomorphism, i.e., it is continuous and invertible, and its inverse is continuous too. Moreover,* $C_O|_{\mathrm{patch}}$ *is open in that case.*

Before we continue to the proof, let us elaborate on the construction of the function $g_O : Z_O|_{\mathrm{patch}} \to C_O|_{\mathrm{patch}}$. In words, the function is constructed by the following scheme:

1. Pick any completion of $z_O$ to $z \in Z|_{\mathrm{patch}}$, and determine the corresponding $y_A$, $\forall A \in P_\mathbb{A}$.

2. Find the corresponding $c$ which gives the same $y_A$ $\forall A$.

3. The $c_O$ in $c$ is the output.

Continuity of $g_O$ is defined with respect to the subset topology of the embedding spaces, i.e., using topologies generated by open $\epsilon$-balls intersected with $C_O|_{\mathrm{patch}}$ and $Z_O|_{\mathrm{patch}}$. Importantly, by the above theorem, $c_O$ is independent of the choice of completion to $z$, i.e. there is a unique $c_O$ corresponding to $z_O$.

*Proof.* Consider any $z_O \in Z_O|_{\mathrm{patch}}$. Using the definitions of $Z_O$ and the $Z_A$, we can first complete $z_O \in Z_O|_{\mathrm{patch}}$ to a $z_A \in Z_A|_{\mathrm{patch}}$ with $A \in \mathcal{A}$, and then complete this $z_A$ to $z \in Z|_{\mathrm{patch}}$. This $z$ will have unique values $y_A$ $\forall A \in P_\mathbb{A}$, given by the bijective $\phi_A^{(z)}$ as $\phi_A^{(z)}(\pi_A(z))$. Similarly, these $y_A$ correspond to a unique $c \in C$ via the bijective maps $\phi_A^{(c)}$, determined by the relation

$$\phi_A^{(c)}(\pi_A^{(c)}(c)) = y_A = \phi_A^{(z)}(\pi_A^{(z)}(z)) \quad \forall A \in P_\mathbb{A}. \tag{6}$$

Eq. (6) fixes all entries of $c$, because the definition of minAIRs requires that for all indices $i$ there exists a $A \in P_\mathbb{A}$ such that $i \in I_A$.

---

[1]Here, the $O$ is for "overlap" and replaces $\mathcal{A}$ in the main text.

Rearranging Eq. (6) gives us

$$\pi_O^{(c)}(c) = \left(\pi_O^{(c)} \circ (\phi_A^{(c)})^{-1} \circ \phi_A^{(z)}\right)(z_A) \tag{7}$$

for all $A \in \mathcal{A}$. For proving that Eq. (5) is well-defined, we have to show that Eq. (7) is independent of the completion $z_O \mapsto z_A$. Furthermore, since the left-hand side does not depend on $A$, Eq. (7) will show that $g_O$ is independent of the choice of $A$.

Now, we prove that Eq. (7) is independent of the choice of completion.

In the following, we make implicit use of Lemma 3 proven below, which makes explicit the connection between a curve and its projection into a subspace. Consider now $z, \tilde{z} \in Z_{\text{patch}}$ with $\tilde{z}_O = z_O$. Since $Z|_{\text{patch}}$ is convex, the straight line which connects $z$ and $\tilde{z}$ while keeping $z_O$ constant is completely contained in $Z|_{\text{patch}}$. This path can be covered with finitely many box-shaped sets. Here, we call a set $B$ *box-shaped* if it is a higher-dimensional closed interval, i.e. $B = \bigtimes_{j=1}^{d_z}[a_j, b_j]$ with $b_j > a_j$.

To find the covering, use the openness to find a closed $\epsilon$-ball $B_1$ with $z$ as center, and a closed $\epsilon$-ball $B_2$ with same $\epsilon > 0$ that has $\tilde{z}$ as center. Because of convexity, the convex hull of $B_1 \cup B_2$ is also fully contained in $Z|_{\text{patch}}$. In particular, every point on the straight line is the center of a closed $\epsilon$-ball that is fully contained in $Z|_{\text{patch}}$ (intuitively, one obtains the convex hull of $B_1 \cup B_2$ by moving the $\epsilon$-ball $B_1$ around $z$ along the straight line towards the $\epsilon$-ball $B_2$ of $\tilde{z}$).

For every $\epsilon$-ball, there exists a $\tilde{\epsilon} > 0$ such that there is a box-shaped subset with side-length $\tilde{\epsilon}$ which has the same center as the ball. On the straight line from $z$ to $\tilde{z}$, consider a lattice with end points $z$ and $\tilde{z}$ and with lattice spacing smaller than $\frac{\tilde{\epsilon}}{10}$. The covering is then given by the $\tilde{\epsilon}$-boxes that have the lattice points as their center.

Inside of each box, we can apply the following argument: Consider any $z^{(1)} \neq z^{(2)}$ in the box with $z_O^{(1)} = z_O = z_O^{(2)}$. Start with $z^{(1)}$, and one by one we exchange an entry of $z^{(1)}$ with that of $z^{(2)}$. Note that this will not make us leave the box. We will now argue that changing one entry in a $\hat{z}$ with $\hat{z}_O = z_O$ in a box will not change the value of the $c_O$ associated to it by Eq. (5).

Within the box, we can change any entry $\hat{z}_i$ with $i \notin I_O^{(z)}$ without changing any of the other entries. That entry is not in the overlap, so there is a $A \in \mathcal{A}$ such that $i \notin I_A^{(z)}$. Since the associated $\hat{z}_A$ does not get changed, Eq. (7), applied to $\hat{z}_A$ instead of $z_A$, tells us that the associated $\hat{c}_O := \left(\pi_O^{(c)} \circ (\phi_A^{(c)})^{-1} \circ \phi_A^{(z)}\right)(\hat{z}_A)$ does not change.

As in every box $c_O$ only depends on $z_O$, and consecutive boxes overlap, this shows us that $c_O$ remains constant along the straight path. Since the endpoints $z$ and $\tilde{z}$ were arbitrary in $Z|_{\text{patch}}$ to begin with (except for having the same $z_O$), this shows that in the entire patch $c_O$ only depends on $z_O$. So we have shown that Eq. (5) is independent of the choice of both completion and $A \in \mathcal{A}$ and therefore well-defined. By definition $g_O$ is then also continuous and surjective.

$C_O|_{\text{patch}}$ is path-connected as the image of a continuous function $g_O$.

The additional assumption $C|_{\text{patch}} \subseteq C_{\text{conv}} \subseteq C$ for a convex and open subset $C_{\text{conv}}$ of $\mathbb{R}^{d_C}$ allows us to repeat the same arguments as in the proof above, but with $z$ and $c$ exchanged, and with $Z|_{\text{patch}}$ replaced by $C_{\text{conv}}$. This gives us a map $g_O^{-1} : \pi_O^{(c)}(C_{\text{conv}}) \to Z_O$. Its restriction to $C_O|_{\text{patch}}$ is then the inverse of $g_O$.

Specifically, starting from any $z_O$ and identifying the corresponding $c_O = g_O(z_O)$, the flipped proof of the theorem shows that this $c_O$ also uniquely determines the $z_O$ which can give the same $y_A \, \forall A$ under any choice of completion of $c_O$ to a $c$.

The openness of $C_O|_{\text{patch}}$ follows from the fact that $C_O|_{\text{patch}}$ is the image of an open set $Z_O|_{\text{patch}}$ under a homeomorphism.

$\square$

**Corollary 1.** *The disentanglement theorem from the main text holds.*

*Proof.* Consider Theorem 2 with $Z|_{\text{patch}} = Z$ and $C_{\text{conv}} = C$. Both are convex by the requirement of the main text theorem and open by definition of minAIRs.

The only part left to be shown is that $C_O|_{\text{patch}} = C_O$, i.e., that the image of $g_O$ is the full set $C_O$.

This is a consequence of the map $g_O^{-1} : \pi_O^{(c)}(C_{\text{conv}}) \equiv C_O \to Z_O$, which together with $g_O : Z_O|_{\text{patch}} \equiv Z_O \to C_O$ shows us that to every $z_O \in Z_O$ there is a unique $c \in C_O$ which can give the same $y_A \; \forall A$ under arbitrary completion of $z_O$ to a $z \in Z$, and vice versa. □

### B.1 Projecting convex curves

Here, we state and prove an illustrative Lemma which is used implicitly in the proof of Theorem 1 to make statements about projected curves.

**Lemma 3.** *Consider an open and convex set $C \subseteq \mathbb{R}^{d_C}$. Pick an index set $I \subset \{1, 2, \ldots, d_C\}$, and use the notation*

$$\pi_I(c) = c_I = (c_i)_{i \in I} \tag{8}$$

*Furthermore, define $C_I = \{\pi_I(c) \; \forall c \in C\}$. Then:*

*$C_I$ is open in $\mathbb{R}^{|I|}$, i.e. each point $c_I \in C_I$ has an open $\epsilon$-ball environment which is a subset of $C_I$. Also, $C_I$ is convex too, which can be elaborated in the following way:*

*For each convex-linear curve in $C$*

$$c(p) = (1-p) \cdot c^{(1)} + p \cdot c^{(2)}, \quad p \in [0,1] \tag{9}$$

*exists a convex-linear curve*

$$\pi_I(c(p)) = (1-p) \cdot \pi_I(c^{(1)}) + p \cdot \pi_I(c^{(2)}), \quad p \in [0,1] \tag{10}$$

*in $C_I$. Similarly, each convex-linear curve $c_I(p)$ in $C_I$ can be lifted to a convex-linear curve in $C$ which satisfies $\pi_I(c(p)) = c_I(p)$.*

*Proof.* Going from Eq. (9) to Eq. (10) works because $\pi_I$ just picks entries corresponding to $I$, but leaves those entries unchanged. By definition of $C_I$, all of the $\pi_I(c(p))$ are in $C_I$. Similarly, going the other way round works because any $c_I^{(1)}, c_I^{(2)}$ can be completed to $c^{(1)}, c^{(2)}$, and the convex-linear curve between them is in $C$ because $C$ is convex.

The openness of $C_I$ follows because any point $c_I$ can be completed to a $c \in C$. Since $C$ is open in $\mathbb{R}^{d_C}$, there is an open $\epsilon$-ball around $c$. Applying $\pi_I$, i.e. leaving out entries which are not $I$, gives us an $\epsilon$-ball in $|I|$ dimensions which is completely contained in $C_I$. □

## C  Alternative definitions of minAIRs

As a motivation for why it may be worthwhile to consider relaxations of the minAIR definition, consider the following example of a pitfall.

**Example 4.** *The following counter-example shows that not all action/experiment settings allow for minAIRs. As a modification of the main text examples, consider three experiments $a = 1, 2, 3$ predicting $y_1 = mg$, $y_2 = qE/m$ and $y_3 = q$, and there are no allowed action combinations, i.e $P_{\mathcal{A}} = \{\{1\}, \{2\}, \{3\}\}$. Since $y_1, y_2, y_3$ are one-dimensional, also their pre-image in the latent-space $z$ should be one-dimensional. Up to permutation, this only gives the option $z = (f_1(m), f_2(q/m), f_3(q))$ for some functions $f_1, f_2, f_3$. However, this breaks the requirement that the full latent space should be open, because $q = \frac{q}{m} \cdot m$. The problem can be traced back to the fact that given $m$ and $q/m$ from experiments $a = 1, 2$, experiment $a = 3$ only reveals redundant information which however has to be computed from several properties already known.*

The example has the problem that it is not always possible to make $Z$ open, while simultaneously making $|I_{A'}|$ minimal such that $\phi_{A'}$ can be invertible. Therefore, one might consider dropping the openness of $Z$, which allows to bring in redundant $z$ entries. Or one might allow the $\phi_A$ to not be homeomorphisms anymore, allowing higher-dimensional $z_A$ to be mapped to lower-dimensional $c_A$. However, we will now discuss counter-examples that show that with such relaxed definitions, we cannot guarantee the correctness of a generalized disentanglement theorem.

**Example 5** ($Z$ *is open, no redundant entries in* $Z$). *In this counter-example, we first make the dimension* $d_Z$ *minimal such that* $Z$ *is open, and then make the dimension of* $|I_{A'}|$ *as small as possible while obeying the first minimization.*

*Consider* $X = (q, m)$ *and four one-dimensional tasks* $y_1 = q$, $y_2 = m$, $y_3 = qm$ *and* $y_4 = q/m$. *By our requirements, we only have two latent nodes, which we pick in the following way:*

$$z = (q, m) \qquad\qquad c = (qm, m) \qquad\qquad (11)$$

*We have* $|I_1^{(z)}| = |I_2^{(z)}| = 1$ *and* $|I_3^{(z)}| = |I_4^{(z)}| = 2$, *while we have* $|I_1^{(c)}| = |I_4^{(c)}| = 2$ *and* $|I_2^{(c)}| = |I_3^{(c)}| = 1$. *This example shows that if we minimize* $d_Z$ *first, there will not be a unique minimal dimension for each* $|I_{A'}|$ *separately. Rather, making one smaller can force another one to become larger. Therefore, a recommended notion of minimality here would be that no* $|I_{A'}|$ *can be made smaller without increasing other ones collectively by at least the same amount.*

*Now, pick task 3 with index sets* $I_3^{(z)} = \{1, 2\}$ *and* $I_3^{(c)} = \{1\}$. *This means* $z_{a=3} = (q, m)$ *and* $c_{a=3} = qm$. *These have different number of degrees of freedom, and can therefore not be bijectively related.*

**Example 6** ($\phi_{A'}$ *are homeomorphisms,* $Z_{A'}$ *minimal.*). *In this counter-example, we will force* $|I_{A'}|$ *to be equal to the dimension of* $Y_{A'}$ *and* $\phi_{A'}$ *homeomorphisms, and then make* $d_Z$ *as small as possible while obeying the first minimization. The counter-example does not allow for action-induced disentanglement.*

*Consider* $X = (x_1, x_2)$ *and one-dimensional tasks* $y_1 = x_1 \cdot x_2$ *and* $y_2 = x_1 + x_2$ *and* **two** *reconstruction tasks* $y_4 = y_3 = (x_1, x_2)$. *Tasks* $a = 1, 2$ *force two of the entries of* $z$ *(up to permutation and application of bijective one-dimensional functions) to be* $z = (x_1 \cdot x_2, x_1 + x_2, \dots)$. *Note that both entries are invariant under a swap* $x_1 \leftrightarrow x_2$. *This means that both entries have the same values for* $(x_1, x_2) = (3, 2)$ *and* $= (2, 3)$. *Therefore, it is* **not** *possible to uniquely determine* $x_1$ *and* $x_2$ *from these two entries alone. But let us consider the following two latent spaces which do allow to solve the reconstruction tasks:*

$$z = (x_1 \cdot x_2, x_1 + x_2, x_1), \qquad\qquad c = (x_1 \cdot x_2, x_1 + x_2, x_2). \qquad\qquad (12)$$

*For the reconstruction tasks, we pick* $I_3 = \{1, 3\}$ *and* $I_4 = \{2, 3\}$, *for both z and c. Then, the overlap nodes of* $A' = \{3\}$ *and* $A'' = \{4\}$ *are just* $z_O = x_1$ *and* $c_O = x_2$, *which are independent degrees of freedom and therefore* **not** *related by a bijective map.*

While the example fits into our framework and satisfies all requirements, we acknowledge that it relies on a bad choice of task-specific latent nodes $I_3$ and $I_4$. Since the two tasks are identical copies, we could just have picked $I_3 = I_4$ instead and no problems would have occurred.

However, this would rely on introducing an extra condition that the trainable mask always picks the latent nodes in such a way that it avoids contradictions like the above. This could be done in a form that says $g_O$ only has to exist for some choices of $I_{A'}^{(z)}$ and $I_{A'}^{(c)}$, rather than particular ones fixed in advance. Such a condition is rather questionable, because it would require already at the definition stage of a latent space $Z$ that the index sets $I_{A'}^{(z)}$ are fixed such that a disentangling map $g_O$ can exist for all other latent representations $c$.

In principle, however, such a unique fixing of index sets could exist if there is one global loss minimum for the trainable mask which uniquely determines the index sets such that they always allow for a disentangling bijective map $g_O$. In our example, the two contradicting tasks are identical, so the existence of such a benevolent global loss minimum seems possible. Nonetheless, in general, it would seem like a very unlikely, fine-tuned coincidence.

# D  Practical applicability of the assumptions leading to Theorem 1

The mathematical assumptions leading to Theorem 1 describe an idealized best case scenario. This raises the question of how close the practical applications of the VAIR architecture will come to satisfying these assumptions. In this section, we will investigate this question in further detail.

The convexity assumption in Theorem 1 is for simplicity of the theorem text. Actually required is that see-saw paths with certain properties are contained in the latent manifold. This is guaranteed by convexity, but convexity is mathematically stronger. Nonetheless, the proof spells out all the assumptions for the see-saw paths and can therefore directly be used to replace the convexity requirement in the theorem.

The openness of the latent manifold in the embedding space is a weaker assumption. Assuming that the noising out of latent neurons manages to make the resulting dimension of the embedding space minimal, the latent manifold will be of full dimension. It might still have a boundary, but this is a measure zero subset that can likely be neglected.

Nonetheless, while the action-conditioned encoder part noises out as many latent neurons as possible, this still allows for redundancy across actions, and with the assumption $d_Z = |\bigcup_A I_A|$ this breaks openness. For example, the latent space could have several copies of the same neuron, and the encoders for different actions decide to keep different copies, while noising out the others. While we did not encounter this problem in the numerics, it can be tackled with a slight modification of the setup. Here, one could make the encoder $E_x$ also output input-independent standard deviations for a typical reconstruction task. Then, one could use the usual ELBO as an additional loss contribution to condition $E_x$ itself to spread the information over as few latent neurons as possible.

In the previous section, we already considered that the invertibility of the decoders and the constraints on the full latent manifold $Z$ can lead to incompatible requirements of minimal dimensions.

The invertibility of the decoders is a strong but conceptually important assumption. It serves the purpose of uniquely identifying the intersection values in one representation with those in another representation. Therefore, the inverse of one decoder is used to define the bijection between the intersections of both representations. If the invertibility does not hold, then one representation could have significantly more redundancy (in the sense of many-to-one) than the other representation.

However, it seems likely that a relaxed theorem would still apply. Assuming that the latent manifolds are still of full dimension in their respective embedding spaces, one could identify one slice or subset of the intersection of one representation with a slice or subset of the intersection of the other representation. Indeed, our generalized theorem in the appendix is already partially prepared for such a situation by allowing to focus on convex subsets, instead of the full latent manifold.

One important obstacle to invertibility may arise if the output dimension of the decoders is smaller than the number of factors of variation required to calculate said output. An appropriate task design can avoid this problem. The strategy is to modify the original task to output more information about the factors of variation.

For example, an originally intended task could be to predict the position $x(t+\Delta t)$ of a particle after one fixed time step, in one spatial dimension and without acceleration. This would require both the position $x(t)$ and the velocity $v$. Therefore, two factors of variation are needed to solve a task with a one-dimensional output, preventing injectivity. However, the task can be modified to also output the velocity $v$, or the previous position $x(t - \Delta t)$. Now, the two factors of variation can be uniquely identified from the task outputs.

The assumption that VAIR perfectly solves the problems is very strong, but required to identify neurons between representations. If the solutions are sub-optimal, it seems likely that maps similar to our bijection still exist, but that they will not be perfectly invertible anymore. This is nonetheless a common assumption in theoretical approaches to VAE and disentanglement.

Continuity is a standard assumption and usually unproblematic. The assumption of continuous manifolds and the polarized regime is also standard in the field of representation learning. Similarly, the assumption

of a unique ground truth is standard in classification and regression, which could be used as tasks. The surjectivity of the polarized encoder $\psi$ is by definition.

We also emphasize that our minAIRs do not assume differentiability, and they achieve disentanglement without the use of Jacobians. The practical VAIR architecture, however, requires differentiability for the sake of a gradient descent optimization.

# E  Datasets

We now present a complete description of the datasets used in the different numerical experiments.

## E.1  Abstract experiment

In Section 4.1 we consider an abstract problem directed to illustrate the various theoretical concepts presented. We define for that a simple dataset arising from four factors of variation $c_1, \ldots, c_4$. An observation is then given by a set of functions of the hidden factors

$$x = \left[ f_k(c_1, \ldots, c_4) \right]_{k=1}^{d_x} \tag{13}$$

where $d_x$ is the observation's dimension and $f_k(\vec{c}) = \sin\left( \sum_i w_i c_i + b_i \right)$ with $w_i$ and $b_i$ being uniformly sampled from $[0, 1]$. We then consider two different actions $a_1$ and $a_2$. These actions (or experiments) produce outputs

$$y_{a_1} = \left[ g_k(c_1, c_2) \right]_{k=1}^{d_y}, \tag{14}$$

$$y_{a_2} = \left[ g_k'(c_2, c_3, c_4) \right]_{k=1}^{d_y}, \tag{15}$$

where $g_k$ and $g_k'$ are functions of the same form as $f_k$. Importantly, the random parameters $w$ and $b$ are fixed prior to training, allowing the models to learn them through training.

## E.2  Classical experiment

**Experiment description**  In Section 4.2 we consider the 2D motion of a particle with mass $m$ and charge $q$ under the influence of two distinct actions. The first, referred to as *the mass action* ($a_M$), considers the elastic collision of the object with a fixed, chargeless test mass $M > m$ with velocity $v_H$, inducing a uniform rectilinear motion along a specified direction $\vec{e}_k$. By means of energy and momentum conservation, and consider an elastic collision, the particle's trajectory after impact is described by

$$\vec{x}_H(t) = \frac{2 M v_H}{M + m} t \cdot \vec{e}_k =: v_m t \cdot \vec{e}_k, \tag{16}$$

The second action, termed *the electric action* ($a_E$), consists of activating a plate capacitor containing the arena where the particle moves. Under the influence of a uniform electric field $\vec{E}$, the particle undergoes a uniformly accelerated rectilinear motion, with the capacitor aligned such that the motion occurs in the $-\vec{e}_k$ direction, i.e. $\vec{E} = E\vec{e}_k$. In this case, one can compute the trajectory of the particle by means of Newton's second law

$$\vec{x}_E(t) = -\frac{qE}{m} t^2 \cdot \vec{e}_k \tag{17}$$

Finally, we extend the analysis to account for the influence of an external magnetic field on the particle's trajectory. For that, we consider that in all experiments the particle has a small initial velocity $v_0$ and consider then the presence of a magnetic field $\vec{B} = B\vec{e}_z$. The resulting motion is uniformly circular and described by

$$\vec{x}_{ext}(t) = R\left[ \cos(\frac{v_0}{R}t + \theta_0)\vec{e}_x + \sin(\frac{v_0}{R}t + \theta_0)\vec{e}_y \right] \tag{18}$$

with the radius $R = \frac{mv_0}{qB}$ and $\theta_0$ being the initial angle of the particle. Importantly, we assume that the particle's acceleration is sufficiently small, allowing the magnetic field contribution to be approximated by Eq. (18), where the velocity is treated as constant and equal to the initial velocity. This approximation avoids the need to solve a differential equation, thereby simplifying the analysis without introducing unnecessary complexity. We then consider that the resulting motion of the particle in the magnetic field, after any of the two actions ($a_E$ or $a_M$) has been applied, is given by $\vec{x}(t) = \vec{x}_{E \text{ or } H} + \alpha_{ext}\vec{x}_{ext}$, where $\alpha_{ext} \in [0,1]$.

**Dataset details** As commented, the observation in this experiment is the concatenation of the trajectories of the particle under both actions. We consider here that the particle is recorded for $T = 200$ time steps, leading to an observation with $d_x = 2 \times 2 \times T = 800$. The action is represented by a one hot encoding of dimension $d_a = 2$, where $[0,1]$ represents the mass action and $[1,0]$ the electric action. The outcome $y$ to be predicted is the trajectory under the given action, hence $d_y = 2 \times T = 400$.

### E.3 Quantum experiment

**Experiment description** In Section 4.3, we consider the quantum tomography of a two-qubit quantum system. We consider that the quantum system is prepared in different arbitrary mixed states $\rho$, a $2^n \times 2^n$ Hermitian matrix, where $n = 2$ is the number of qubits. We then select 75 random measurements $\{|\psi_j\rangle \langle\psi_j|\}_{j=1}^{75}$, fixed before training, that provide a collection of observations, i.e., the probability of finding the state $\rho$ in any of the states $|\psi_j\rangle$, $x = \{(p_j)_{j=1}^{75}\big| \, p_j = \langle\psi_j| \rho |\psi_j\rangle \in [0,1]\}$. Then we select Pauli projectors of the form $w_{i,k} = (\mathbb{I} + \hat{\sigma}_i^{\text{I}} \otimes \hat{\sigma}_k^{\text{II}})/2$, where $i,k = 0,...,3$ point to the different Pauli matrices

$$\hat{\sigma}_x = \begin{pmatrix} 0 & 1 \\ 1 & 0 \end{pmatrix}, \quad \hat{\sigma}_y = \begin{pmatrix} 0 & -i \\ i & 0 \end{pmatrix}, \quad \hat{\sigma}_z = \begin{pmatrix} 1 & 0 \\ 0 & -1 \end{pmatrix}, \quad \mathbb{I} = \begin{pmatrix} 1 & 0 \\ 0 & 1 \end{pmatrix} \tag{19}$$

In particular, $\hat{\sigma}_0 = \mathbb{I}, \hat{\sigma}_1 = \hat{\sigma}_x, \hat{\sigma}_2 = \hat{\sigma}_y, \hat{\sigma}_3 = \hat{\sigma}_z$, which form a basis of Hermitian operators. Moreover, the superindices (I), (II) denote the qubit indices (1) and (2), respectively. The set of projectors $\{w_{i,k}\}_{i,k=0}^{3}$ contains only non-trivial projectors which excludes $w_{0,0} = \mathbb{I}$ and thus $|\{w_{i,k}\}_{i,k=0}^{3}| = 15$. The goal is to predict the expectation value of a selected Pauli projector $a_{i,k} = w_{i,k}$, $y_{a_{i,k}} = \text{tr}(w_{i,k}\rho)$.

We further create a test set consisting of states of the form $\rho(r) = \frac{1}{4}\left(\mathbb{I} + r(\hat{\sigma}_i^{\text{I}} \otimes \hat{\sigma}_k^{\text{II}})\right)$, with $r \in [-1,1]$, to probe the learned representation. Since the encoder $E_x$ acts directly on the state $\rho$, its output $\mu$ is expected to depend on the parameter $r$, as demonstrated in the lower panels of Figure 4b. Each latent variable $\mu_l$, corresponding to one of the 15 latent neurons of $E_x$, responds exclusively to a specific operator $\hat{\sigma}_i \otimes \hat{\sigma}_k$. This behavior indicates that the VAIR has accurately learned the Bloch representation of the two-qubit state Gamel (2016).

**Dataset details** As described above, the input to the encoder $E_x$ is a vector of dimension $d_x = 75$, corresponding to random projective measurement outcomes obtained from the density matrix $\rho$. The input to the second encoder, $E_a$, consists of the actions $a_{i,k}$, represented by $4 \times 4$ complex Pauli projectors. To process these inputs, each complex matrix is first flattened into a column-vector representation. The real and imaginary components are then separated to form two real-valued vectors of equal length, which are subsequently concatenated. This procedure yields a real-valued input vector of dimension $d_a = 2 \cdot 4 \cdot 4$. Finally, the decoder outputs a single measurement outcome $\text{tr}(w_{i,k}\rho)$ with dimension $d_y = 1$.

## F   Training VAIRs

In this section, we provide an overview of the variational autoencoder architecture used throughout this work. For all experiments presented in the main text, we use the same architecture. However, the dimensions of the observations $d_x$, actions representations $d_a$ and output $d_y$ vary, as commented above and presented in Table 1. Moreover, we also provide there further details such as the dataset size and the batch size used during training.

| Experiment | $d_x$ | $d_a$ | dim $Z$ | $d_y$ |
|---|---|---|---|---|
| Abstract (Section 4.1) | 10 | 2 | 4 | 10 |
| Classical Physics (Section 4.2) | 800 | 2 | 2 | 400 |
| Quantum Physics (Section 4.3) | 75 | 32 | 15 | 1 |

Table 1: **Dataset dimensions.** Summary of the dimensions of the datasets used for training VAIR and other VAE variants. dim $Z$ refers here to the dimension of the factors of variation (see Section 2).

### F.1 Architecture

The ML model used in this work is schematically represented in Figure 1d. A detailed description of the neural network used throughout the work is presented in Table 2. As mentioned in the main text, it consists of two encoder networks, $E_x$ and $E_a$. For this work, both have a similar architecture, although that is not a requirement. On the one hand, $E_x$ gets as input the observation $x$ and predicts the mean values $\mu_i$ for each latent neuron $z_i$. On the other hand, $E_a$ gets as input the action $a$ and predicts the variances' logarithm $\log \sigma_i^2$ of the latent neurons. The latent space, composed of $d_z$ latent neurons, is then sampled using a multivariate normal distribution. In particular, each latent neuron's value $z_i$ is sampled from $\mathcal{N}(\mu_i, \sigma_i)$. During training, we use the reparameterization trick to allow for gradient calculation, so that $z_i = \mu_i + \sigma_i \epsilon$, where $\epsilon$ is normal noise, as typically done in VAEs. The sampled latent space is then concatenated to the action representation and fed into the decoder $D$, which then predicts the outcome $y$.

| Module | Layers |
|---|---|
| **Encoder** $E_x$ | Linear($d_x$, 512) |
| | Relu |
| | Linear(512, 256) |
| | Relu |
| | Linear(256, $d_z$) |
| **Encoder** $E_a$ | Linear($d_a$, 512) |
| | Relu |
| | Linear(512, 256) |
| | Relu |
| | Linear(256, $d_z$) |
| **Decoder** $D$ | Linear($d_z + d_a$, 256) |
| | Relu |
| | Linear(256, 512) |
| | Relu |
| | Linear(512, $d_y$) |

Table 2: **VAIR architecture.** Layer composition of the different modules of the VAIR architecture

### F.2 Training details

In Table 3 we present the training hyperparameters used in the different experiments. We use in all cases the Adam optimizer with gradient moving average coefficient $\beta_1 = 0.9$ and gradient squared moving average coefficient $\beta_2 = 0.99$. The training is performed by minimizing the ELBO in Equation (4), where the reconstruction loss (first term) follows the usual choice of the mean squared error:

$$\mathcal{L}_R = \frac{1}{N} \sum_{i=1}^{N} (y_i - y_i')^2 \tag{20}$$

where $y$ is the ground truth output and $y'$ is the prediction of the decoder $D$.

| Experiment | Dataset size | Batch size | Epochs | Learning rate | $\beta$ | $d_z$ |
|---|---|---|---|---|---|---|
| Abstract (Section 4.1) | $6 \cdot 10^3$ | 100 | 5000 | $10^{-4}$ | $10^{-2}$ | 6 |
| Classical Physics (Section 4.2) | $10^4$ | 50 | 200 | $10^{-4}$ | $10^{-3}$ | 2 |
| Quantum Physics (Section 4.3) | $6 \cdot 10^3$ | 200 | 2000 | $10^{-4}$ | $10^{-3}$ | 15 |

Table 3: **VAIR hyperparameters.** Training hyperparameters for the training of VAIR in different experiments

## G   VAE benchmarks

### G.1   Architectures

In Section 4.1, we compared the performance of VAIR against four VAE-based baselines on the abstract experiment dataset. Figure 5 provides a schematic overview of these architectures.

The first variant, TC-VAE$_x$, is a standard VAE trained to reconstruct the observation $x$ from itself, i.e., both the input and output are $x$. In principle, this should suffice to recover the hidden variables of the system. To promote disentanglement, we train the model using the Total Correlation (TC) loss from Ref. Chen et al. (2018), defined as:

$$\mathcal{L}_{\text{TC}} := \mathbb{E}_{q(z|x^{(n)})p(x^{(n)})} \left[\log p(x^{(n)} \mid z)\right] - \alpha_{\text{TC}} \, I(z; x^{(n)}) - \beta_{\text{TC}} \, \text{KL}\left(q(z) \, \| \, \prod_{j=1}^{d_z} q(z_j)\right) - \gamma_{\text{TC}} \sum_{j=1}^{d_z} \text{KL}(q(z_j) \, \| \, p(z_j)), \tag{21}$$

where $I(z; x^{(n)})$ denotes the mutual information between latent and input.

The second variant, VAE$_{x,a}$, is a $\beta$VAE that receives both the observation $x$ and the action $a$ as input and is trained to predict the experimental outcome $y$. The third model, VAE$_{D_a}$, shares the same input structure as VAE$_{x,a}$, but also feeds the action $a$ directly into the decoder—replicating the decoding path of VAIR. Finally, the fourth variant, TC-VAE$_{D_a}$, uses the same architecture as VAE$_{D_a}$ but is trained with the TC loss described above.

To make the comparison as fair as possible, we devise the encoders and decoders of these architectures to have approximately the same number of parameters of the VAIR presented in Table 2 (see Table 4 and Table 5). In all cases we train with $z = 6$. The training then proceeds in the same way as for the latter, see Appendix F.2 for details.

In Figure 6, we show the loss in Equation (4) over training, separated in its two components: the reconstruction and latent terms. As stated in Appendix F.2, the former is parameterized by a mean squared error loss (MSE), while the second one computes the Kullback-Leibler divergence between the latent neurons and the prior $\mathcal{N}(0, 1)$. The higher MSE of VAE$_x$ is expected: this model is trained to reconstruct the observation $x$ rather than the action-dependent output $y_A$, making the problem inherently different and the MSE values not directly comparable to the other models. For all remaining models, the reconstruction losses are in close agreement, confirming that the observed differences in disentanglement are not due to tradeoffs with reconstruction quality, but rather reflect genuine differences in the learned latent representations. The small variations in KL divergence across models are consistent with the minor differences in each model's architecture and training objective

### G.2   Learned representations and mutual information gap (MIG)

We now explore the representations learned by the aforementioned VAE variants. For TC-VAE$_x$, no actions are considered, hence the model relies on the inductive bias of the TC-loss for disentanglement. On the other hand, VAE$_{x,a}$ must encode information about the action in the decoder, as the decoder needs it to reconstruct $y_a$. This rules out the AIR paradigm for this architecture. On the other hand, both VAE$_{D_a}$ and TC-VAE$_{D_a}$ can, in principle, reach minAIRs, as the action is fed into the decoder.

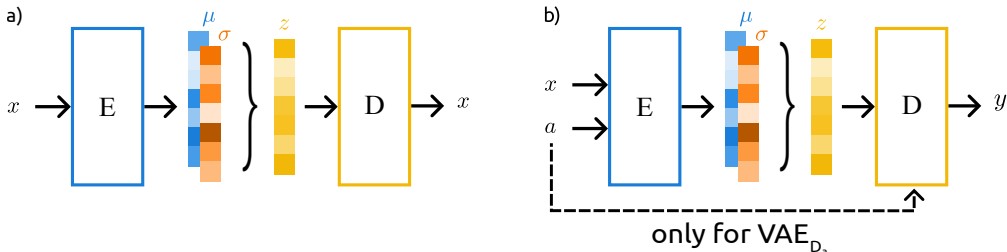

Figure 5: **Schematic representation of VAE benchmark architectures** a) TC-VAE$_x$, having as input and output the observation $x$. b) VAE$_{x,a}$ having as input the observation $x$ and $a$, and predicting the output $y$. Moreover, we further consider two variants, VAE$_{D_a}$ and TC-VAE$_{D_a}$, where the action is also input to the decoder.

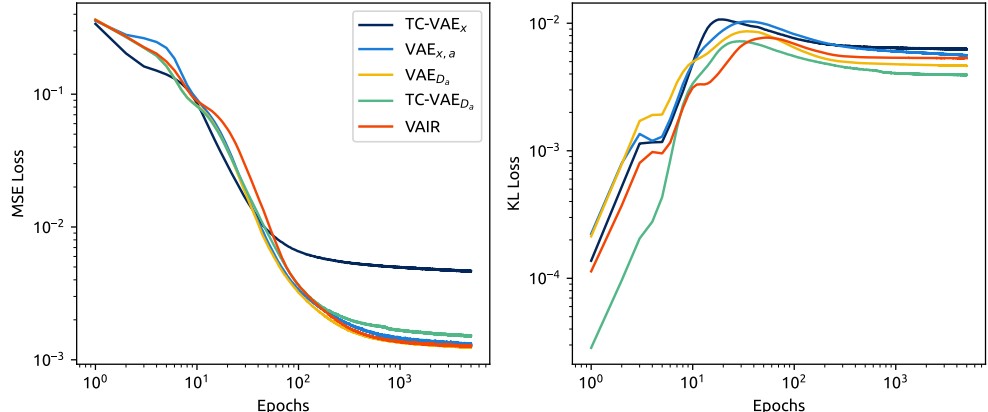

Figure 6: **Training losses for VAE benchmarks architectures and VAIR** We compare here the two terms of the loss Equation (4), namely the reconstruction loss (left), here the MSE, and the latent loss (right), here the KL loss.

To analyze this behavior, we consider the abstract dataset of Section 4.1 and compute the mutual information gap (MIG) for each hidden variable $c_i$, calculated via the mutual information between $c_i$ and both the mean of the latent neurons $\mu_k$ (Figure 7a,b) and the sampled value $z_k$ (Figure 7c,d). The former, which we refer to as MIG$_\mu$, gives a more interpretable visualization of what is really encoded in each latent neuron and hence is then one shown in the main text in Figure 2. For completion, we also provide the latter, which we refer to as MIG$_z$, for which the variance $\sigma_k^2$ is also considered.

In Figure 7 we show both MIGs for each input action $a_i$ for each of the following architectures: VAE$_{x,a}$, VAE$_{D_a}$, TC-VAE$_{D_a}$ and VAIR. Focusing first on MIG$_\mu$ (Figure 7a,b), we see no change for VAIR w.r.t. the action, as the means $\mu_k$ depend solely on the observation $x$. On the other hand, we see a strong dependence for the rest of the models, meaning a correlation between the action and the representation. In particular, for all three benchmark models the MIG$_\mu$ for $c_{3,4}$ vanishes for action $a_1$. Indeed, these two variables are not needed for $y_{a_1}$ and hence the models completely disregard these. The same is applicable for $c_1$ and action $a_2$. When considering MIG$_z$ (Figure 7c,d), the variances enter into play, and we see now that also the MIG for VAIR follows the same pattern, as they are mixed latent neurons and are noised out when not required for the given action. We also see an expected decrease in the MIG due to the noise.

Another important takeaway from this is the fact that, when considering the effect of actions in Figure 7b,d), the TC-VAE$_{D_a}$ has a higher average MIG for all hidden variables but $c_2$. This is to be expected, given the results of Theorem 1. Indeed, one could train VAIR in combination with the TC-loss to further improve the overall disentanglement. However, as presented in Section 4.2, in some cases there exist natural, interpretable representations that emerge in VAIR (i.e., $z_1 = m$ and $z_2 = q/m$) that go against the objective of the TC-loss.

| Module | Layers |
|---|---|
| **Encoder** $E_x$ | Linear($d_x$ / $d_x + d_a$, 1000) |
| | Relu |
| | Linear(1000, 256) |
| | Relu |
| | Linear(256, $2d_z$) |
| **Decoder** $D$ | Linear($d_z+d_a$, 260 / 256) |
| | Relu |
| | Linear(260/ 256, 512) |
| | Relu |
| | Linear(512, $d_x$ / $d_y$) |

Table 4: **Benchmark architectures** Layer composition of the encoder and decoder for the four VAE architectures used for benchmarking. Green highlight properties unique of TC-VAE$_x$ and red of VAE$_{D_a}$ and TC-VAE$_{D_a}$.

| Model | Number of parameters | $\beta$ | $\alpha_{\text{TC}}$ | $\beta_{\text{TC}}$ | $\gamma_{\text{TC}}$ |
|---|---|---|---|---|---|
| TC-VAE$_x$ | 410922 | - | $1.45 \times 10^{-4}$ | $4.25 \times 10^{-3}$ | $5.36 \times 10^{-4}$ |
| VAE$_{x,a}$ | 412922 | $10^{-3}$ | - | - | - |
| VAE$_{D_a}$ | 411358 | $10^{-3}$ | - | - | - |
| TC-VAE$_{D_a}$ | 411358 | - | $7.01 \times 10^{-4}$ | $3.73 \times 10^{-3}$ | $1.00 \times 10^{-4}$ |

Table 5: **Benchmark hyperparameters** Training hyperparameters for the benchmark models. All models, including VAIR (see Table 3) were trained for 5000 epochs and a learning rate of $10^{-4}$.

## H  Large version of the abstract experiment

In this section we replicate the abstract experiment of Section 4.1 with $d_x = 500$, $d_a = 18$, dim$Z = 10$ and $d_y = 500$. Moreover, we consider now four different actions and the following actions-outcome functions:

$$y_{a_1} = G_1(c_1, c_2, c_3) \tag{22}$$
$$y_{a_2} = G_2(c_3, c_4, c_5) \tag{23}$$
$$y_{a_3} = G_3(c_6, c_7, c_8) \tag{24}$$
$$y_{a_4} = G_4(c_8, c_9, c_{10}) \tag{25}$$
$$\tag{26}$$

From the previous, one can identify two intersections between actions, at $c_3$ and $c_8$. Following the results shown in Section 4.1, these are the two factors that will be disentangled.

To showcase this, we train VAIR and TC-VAE$_{D_a}$ on this dataset. We maintain for both the same architectures and hyperparameters presented in Table 3 and Table 5, although we scale the latent space to $d_z = 20$. In Figure 8 we show the MIG for each factor, replicating what was shown in Figure 2. Focusing on the MIG computed with the latent means $\mu$ over the full validation dataset (leftmost plot in Figure 8), we see that VAIR again retrieves high MIGs for the intersection. Conversely, the MIG for TC-VAE$_{D_a}$ is surprisingly low when compared to previous cases. This is because this model finds a representation in which only three neurons encode the factors' information (see Figure 9). As the action changes, the model encodes different factors of variation in those three neurons. This means that the representation *depends* on the action, a condition shown to prevent AIR. When considering the action-dependent MIG (four rightmost plots in Figure 8), we see that the MIG of TC-VAE$_{D_a}$ indeed increases, but is never higher than that of VAIR for the factors in the actions' intersections.

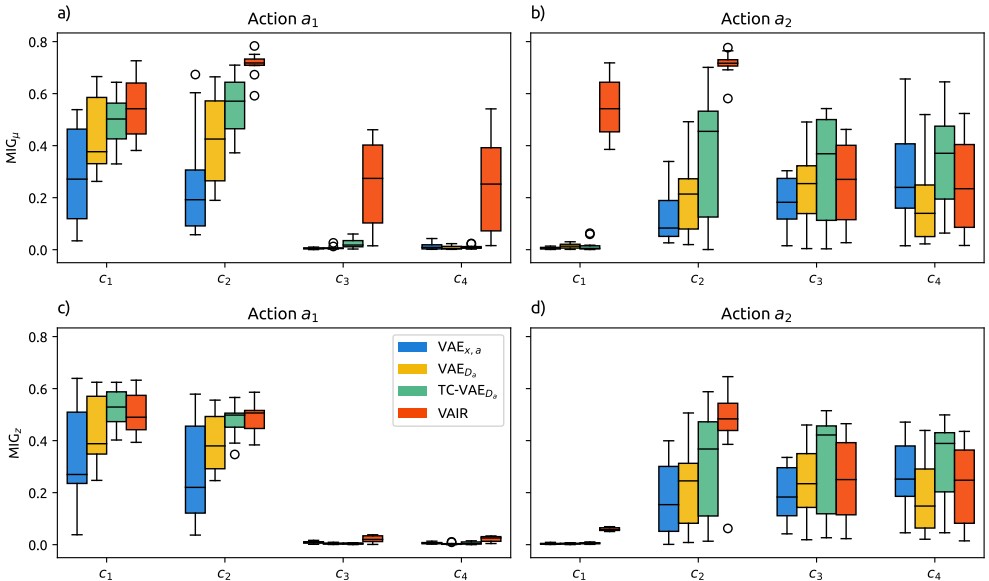

Figure 7: **Comparison of action depending MIG** For the abstract experiment, we compare the MIG for each actions, when computed from the mean $\mu$ of the latent neurons (upper row) and the sampled latent vector $z$ (lower row). Boxplots show results from 20 independent training runs per model, each with random dataset generation and weight initialization.

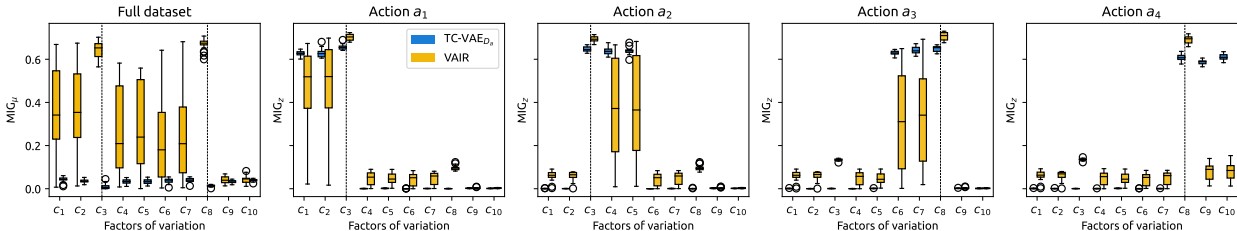

Figure 8: **MIG analysis for large abstract dataset** For the large abstract experiment in Appendix H, the leftmost plot shows the MIG computed over the full validation dataset, while the subsequent plots show the MIG for each input action. The MIG in the first case is computed from the means $\mu$, while the latter is computed from the sampled latent vector $z$. Boxplots show the results from 20 independent training runs per model, each with random dataset generation and weight initialization. Vertical dashed lines show the latent factors in the intersections. Note that $c_3$ is only used for actions $a_{1,2}$ and $c_8$ is only used for actions $a_{3,4}$

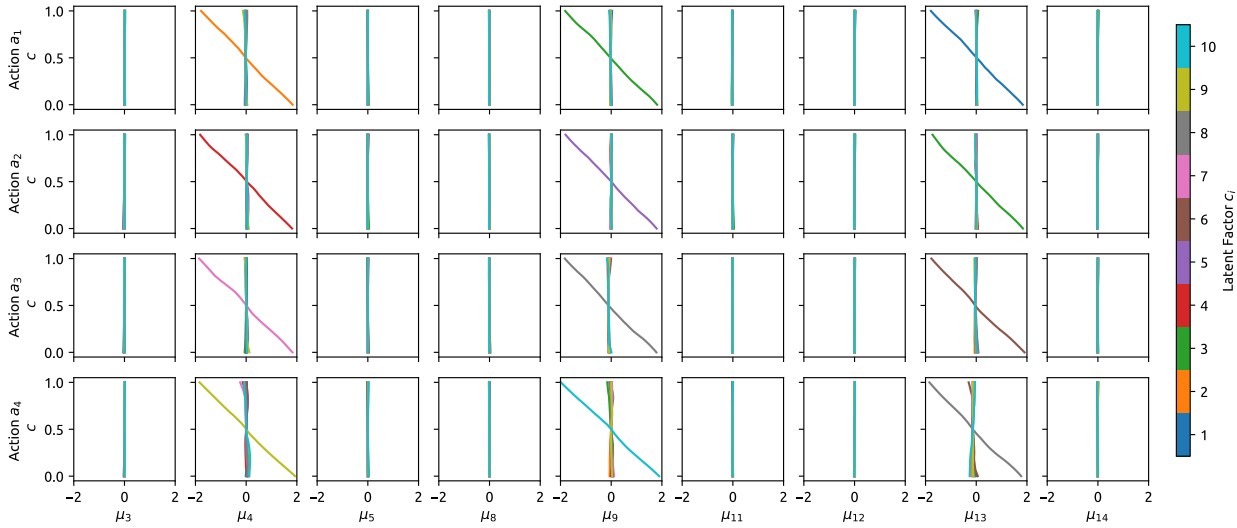

Figure 9: **Learned TC-VAE**$_{D_a}$ **for the large abstract experiment.** Each row shows the latent representation for each action $a_i$ as a function of the value of the given factor of variation $c_i$. The model has 20 neurons. Those not plotted are passive and have the same behavior as $\mu_3, \mu_5, \mu_8, \ldots$; for presentation purposes, they are omitted from this figure.

