# OpenReview forum: "Disentanglement by means of action-induced representations"
_TMLR — Under review for TMLR_

### Review · Reviewer_im1E · 2026-06-24

**Summary Of Contributions:**

This paper introduces Action Induced Representations (AIR) which is a framework to learn disentanglement represntations from action-observations pairs instead of observations only. The main contribution is built on top of minAIR and its assumptions: factors that are shared across actions can be identified up to invertible transformations. Then the authors use the theorem 1 to built VAIR which uses a VAE-like framework to approximate these representations.

The paper is well motivated and the theoretical framework is interesting. Examples for the theory are very helpful to understanding the intuition behind AIR. Also, experiments show some demonstrations of the method.

However, the theoretical guarantees aren't fully meet in VAIR and maybe this needs to be explicitly explained since it leaves that impression that VAIR supports minAIR.

**Audience:**

Yes

**Audience Explanation:**

I expect the work to be of interest to researchers studying representation learning and identifiability, as well as those interested in learning interpretable representations in scientific and experimental settings. The framework itself is novel, and the physics-inspired examples help illustrate how action information can be used to recover meaningful latent structure.

**Broader Impact Concerns:**

I do not have significant broader impact concerns regarding this work.

**Claims And Evidence:**

Yes

**Claims Explanation:**

The theoretical claims are supported with examples, definitions and proofs. In particular theorem 1 is put under clearly stated assumptions. And the empirical results are consistent with the theoretical framework and demonstrate that VAIR can recover action dependent factors in several controlled settings. The abstract experiment include comparisons with multiple VAE baselines and--including physics and quantum experiments--provide qualitative and quantitative evidence that the proposed architecture captures the intended structure.

My main reservation concerns the connection between the theoretical framework and the practical model. Theorem 1 proves disentanglement for minAIRs under the assumptions of Definition 3, while Appendix D discusses how these assumptions may approximately arise in VAIR. In particular, Definition 3 requires invertible task-specific decoders $\phi_A$, whereas VAIR employs a neural-network decoder trained solely through the ELBO objective (Eq. 4). The discussion in Section 3 and Appendix D provides useful intuition for why VAIR may approximate minAIRs, but the link remains heuristic rather than formal. I therefore encourage the authors to more explicitly separate the theoretical guarantees established for minAIRs from the empirical behavior observed for VAIR.

**Requested Changes:**

Critical:
- Clarify more explicitly throughout the paper that Theorem 1 establishes guarantees for minAIRs under the assumptions of Definition 3, while VAIR is a practical architecture motivated to approximate these representations.
- Expand the discussion connecting the assumptions required by Theorem 1 (specifically invertibility of the task specific decoder $\phi_A$ and the latent-space assumptions) to the practical VAIR implementation.
- I would be keen to see how iVAE performs on the abstract experiment. As discussed, iVAE conditions the prior $p(z|u)$, whereas VAIR conditions the posterior through the action-dependent encoder $q(z|x,a)$. Given this, and the fact that actions could naturally be used as auxiliary variables, iVAE seems like a particularly strong baseline alongside VAED.

Non-Critical:
- Provide additional discussion of failure cases, such as situations where actions do not sufficiently isolate factors of variation.
- Consider including additional intuition for the openness and invertibility assumptions in the main text, as these assumptions are central to the theoretical development but may be difficult for readers unfamiliar with the underlying topology concepts (maybe just expand a little bit more Appendix A: a short intuitive example might improve readability).
- The classical physics example provides an intuitive illustration of action dependent representations, the evaluation is primarily qualitative. Given the low-dimensional latent space, it is difficult to assess disentanglement beyond visual inspection. Quantitative analyses similar to those used in the abstract experiment would further strengthen the empirical results.

---

> ### Author Response · Authors · 2026-07-15
>
> We thank the reviewer for their positive comments on our manuscript. Following their comments and those of the other reviewers, we have now better connected the theoretical guarantees of AIR with the VAIR architecture. We comment on each requested change (RC) below.
>
> **RC1 and RC2: Connection between Theorem 1 and VAIR, and its assumptions** Following this comment and those of the other reviewers, we added remarks in several places to emphasize that VAIR seeks to approximate minAIRs, but is not itself a guaranteed minAIR. These connections, and the validity of the assumptions of Theorem 1 in a practical setup, were already discussed in Appendix D; we now bring part of that discussion into the main text, both in the abstract and in the body, to ensure the reader has a fair picture of the framework. In particular, at the end of the *”Why does VAIR approximate minAIRs?”* subsection, we now explicitly reference Appendix C, which contains the problematic examples arising when one of the minimality assumptions fails, and Appendix D, which discusses how closely VAIR comes to producing minAIRs. The main text briefly summarizes this discussion: revealing the actual factors of variation requires many, carefully designed experiments.
>
> **RC3: iVAE comparison** While iVAE bears some resemblance to the VAIR architecture, the structure of the data on which the two are trained, and their objectives, largely differ. iVAE assumes that the data arises from a conditional distribution $p(z|u)$, in which the latent factors $z$ are conditioned on an auxiliary variable $u$. This is fundamentally different from the scenario considered here, where the latent factors are sampled independently of the action $a$ that will be performed, i.e. $p(z|a) = p(z)$. Since iVAE's identifiability relies precisely on the variability of the prior across values of $u$, and no such variability exists in our setting, its identifiability statements do not apply to the AIR scenario.
>
> We now explain these differences more clearly in the Related Work section.
>
> Following the reviewer's suggestion, we nonetheless trained an iVAE on the abstract experiment using the actions as auxiliary variables. As expected from the above, it fails to disentangle the factors of variation. We have therefore not included it in the benchmark of Sec. 4.1, as its underlying assumptions are not met in the AIR setting and the comparison would not be informative about the relative merits of the two approaches.
>
> **RC4: Failure cases** We agree that it is important to inform readers about the cases in which AIR fails to disentangle. To that end, we have extended the text after Example 3, explaining when actions do not isolate factors of variation. In particular, this happens for factors lying outside any intersection, and when an intersection contains multiple factors, since the theorem only guarantees disentanglement between the factors inside and outside the action intersections.
>
> **RC5: Openness and invertibility conditions** Following the reviewer's comment, we have extended Appendix A to better describe the openness and invertibility assumptions. We have also included a condensed version of these descriptions just after the definition of minimal AIRs.
>
> **RC6: Quantitative analysis in the classical experiment** Our aim with this example was to show how one can leverage VAIR to gain knowledge about a real-world physical problem (in the same spirit as Sec. 4.3). We therefore do not aim to benchmark VAIR against a metric, but rather to show what type of representations one may expect from VAIR, and how a prospective user can work with the model. We nonetheless note that the variances in Fig. 3b and c are averaged over 20 different models, showcasing the robustness of the method against different initializations. This was not properly acknowledged in the text, which weakened the presented result; we now report it in the caption of Fig. 3 and in the main text.

---

### Review · Reviewer_M4Yi · 2026-06-26

**Summary Of Contributions:**

The paper proposes action-induced representations (AIRs). A system has latent factors $c$, observed as state $x$; a set of allowed actions $A$ each yields a *deterministic* outcome $y_A(x)$ depending only on a factor subset $I_A$. A minimal AIR (minAIR, Def. 3) requires an encoder $\psi$, per-action invertible decoders $\varphi_A$, and $d_Z = |\bigcup_A I_A|$. Theorem 1/2 shows that factors in the intersection $I_{\mathcal{A}} = \bigcap_{A \in \mathcal{A}} I_A$ of an action set are provably disentangled — disentanglement is tied to the *intersection structure* of the action set. VAIR (Sec. 3) is a VAE realization with a second encoder $E_A$ producing action-dependent log-variances that gate irrelevant neurons. Experiments: an abstract 4-factor benchmark (20 seeds, MIG/modularity vs. TC-VAE/VAE variants), a classical-physics particle setup, and a 2-qubit tomography demo.

Strengths
- The intersection-structure characterization: which factors are identifiable is the intersection $I_{\mathcal{A}} = \bigcap_A I_A$ of the actions' dependency sets, is a clean, genuinely interesting idea.
- The abstract experiment (Sec. 4.1) is methodologically sound: 20 seeds, matched reconstruction loss, sensible baselines.

Weaknesses
- The identifiability is produced by the per-action invertibility assumption on $\varphi_A$, *not* by introducing actions, so the headline thesis is mis-attributed (C1).
- That enabling assumption (bijective $\varphi_A$, hence $d_{Y_A} \ge |I_A|$) is strong and, by the paper's own examples, routinely violated (C2).
- "Provable disentanglement" is shown for idealized minAIRs but only *heuristically* for the trained VAIR (C4).
- Observational-vs-interventional data framing is inconsistent and the paired-data requirement is understated (C3).
- Closest related work (causal representation learning, JEPA, world models) is missing (see Audience).

**Audience:**

No

**Audience Explanation:**

As *stated*, the findings would mislead a reader into believing actions are a general route to provable disentanglement. The positioning is also incomplete on two fronts an interested reader would immediately raise:
- JEPA and the world model thread: the "predict outcome from state+action" skeleton overlaps with joint-embedding predictive architectures, which are neither cited nor contrasted (VAIR does differ: generative decoder + ELBO + action-conditioned variance gating).
- Causal representation learning: the closest competitors are missing. In particular Li, Pan & Bareinboim, "Disentangled Representation Learning in Non-Markovian Causal Systems" (NeurIPS 2024), whose *"algorithm that returns a causal disentanglement map, highlighting which latent variables can be disentangled"* is the direct analogue of the $I_A$/$I_{\mathcal{A}}$ mechanism but achieved under unobserved confounding and multi-domain data *without* a deterministic bijective map; and Mao, Xia et al., "Causal Transportability for Visual Recognition" (CVPR 2022), which recovers domain-invariant causal representations *"using representations in deep models as proxies."* Both sharpen C1–C2: the restrictive assumptions look like choices, not necessities.

**Claims And Evidence:**

No

**Claims Explanation:**

Verdict: the central claims are not supported as stated. The mathematics appears internally correct (I checked the proof of Thm 2, App. B), but it establishes something weaker and different from the abstract.

- C1: Identifiability comes from the invertibility assumption, not from "actions" (the core issue). The disentanglement map $g_{\mathcal{A}} = \pi_{\mathcal{A}}^{(c)} \circ (\varphi_A^{(c)})^{-1} \circ \varphi_A^{(z)}$ (Eq. 3) is well-defined *only because* $(\varphi_A)^{-1}$ exists (Def. 3, cond. 1). Once each action's outcome map is assumed bijective onto the factors it touches, those factors are recoverable *by assumption*; the action set merely determines *which* subsets/intersections are isolated. This is structurally the standard lever of identifiable-representation learning (iVAE; multi-view identifiability, von Kügelgen et al.). So the paper's thesis that *introducing actions* yields provable disentanglement where VAEs fail, mis-attributes the source of the result. The honest claim is: "given invertible/minimal per-action outcome maps, the action set's intersection structure characterizes disentanglement."

- C2: The enabling bijectivity/determinism assumption is strong and routinely violated. It requires each action's outcome to carry at least as much information as the factors it depends on ($d_{Y_A} \ge |I_A|$, injective). When the outcome is lower-dimensional, distinct factor configurations (or distinct actions) collapse to the same outcome and $\varphi_A$ is non-invertible (one can easily find many real life examples where two different sets of state + action lead to the same outcome). The paper's own Appendix C (Examples 4–6) and Appendix D (the 1-D $x(t+\Delta t)$ case: "two factors... one-dimensional output, preventing injectivity") show this is the common case, "fixable" only by redesigning the measurement. The candor is commendable but does not rescue an abstract that advertises "provable disentanglement" without this qualifier.

- C3: Observational vs. experimental data. P. 1 calls the passive VAE baseline "a collection of experimental measurements" backwards; that regime is *observational*. Substantively, Def. 1 requires paired $(x, y_A(x))$ for every action applied to the same state, which is *interventional* data and stronger than "a fixed dataset of observations." The data-access model should be stated plainly and reconciled with the single-observation recoverability asserted in Sec. 4.2.

- C4: Proven for minAIRs, only heuristic for VAIR. Thm 1/2 assumes convex $Z$/$C$ and bijective $\varphi_A$; Sec. 3 admits the VAIR$\leftrightarrow$minAIR correspondence is "heuristic." Calling VAIR's disentanglement *provable* (Abstract) overstates what is shown.

- C5: Experiments are uneven. The abstract experiment (4.1, 20 seeds, matched reconstruction loss) is sound and honestly reported. The physics/quantum experiments (4.2/4.3) appear to be single/few runs with no seed counts or error bars at parity.

**Requested Changes:**

1. Re-attribute identifiability to the invertibility/minimality assumptions (addresses C1); restate the contribution as the intersection characterization *conditional on* those assumptions, and position against iVAE / multi-view identifiability.
2. Scope the "provable" claim and state the bijectivity assumption in the main text (C2): make explicit that Thm 1 needs each $\varphi_A$ bijective onto $I_A$ ($d_{Y_A} \ge |I_A|$), and surface the App. C/D failure cases. Qualify the abstract.
3. Fix the observational-vs-experimental framing and state the paired-data requirement (C3).
4. Separate "proven for minAIRs" from "approximated by VAIR" in the abstract and Sec. 3 (C4).
5. Add and contrast the missing literature: The two causal papers and JEPA, world models / model-based RL such as Dreamer, replacing the unsupported "fundamentally different from CRL" assertion with a precise comparison to intervention-subset identifiability.

Minor: report seeds/error bars for 4.2/4.3 at parity with 4.1; disambiguate the symbol $Z$ (latent space vs. number of factors, Table 1); fix the duplicated Figure 7 caption.

---

> ### Author Response · Authors · 2026-07-15
>
> We thank the reviewer for their very thorough review of our manuscript, and in particular for their careful reading of our theoretical contribution. We acknowledge that several statements could indeed be improved, which we have now done following the reviewer's guidance.
>
> **Requested change (RC) 1.1: Re-attribute identifiability to the invertibility/minimality assumptions** We agree that the dependence of our results on the invertibility and minimality assumptions was not sufficiently stressed in parts of the main text, and we have now brought the relevant clarifications from App. A, C and D into both the abstract and the body, so that the provable identifiability is presented together with the assumptions that enable it.
>
> We would like to respectfully push back, however, on the attribution itself. Invertibility of each $\phi_A$ is *necessary* for our result, but it is not what produces disentanglement: a single action with bijective $\phi_A$ determines the block $z_A$ only up to an arbitrary reparameterization, which is exactly the nonlinear-ICA indeterminacy we cannot escape. What isolates *individual* factors is the intersection structure of the action set. This is made explicit in Example 3: the bijectivity assumptions are identical in both cases, yet $\mathbb{P}_A=\{\{1\},\{2\},\{1,2\}\}$ disentangles both $m$ and $q$, while $\mathbb{P}_A=\{\{1\},\{1,2\}\}$ disentangles only $m$. Since the assumption is unchanged and the disentanglement is not, the assumption alone cannot be its source. We therefore believe the formulation suggested by the reviewer ("given invertible/minimal per-action outcome maps, the action set's intersection structure characterizes disentanglement") is an accurate description of our contribution, and we have adopted this framing in the revised text.
>
>
> **RC1.2: Relation to iVAE and multi-view identifiability** We also thank the reviewer for highlighting important work on causal and multi-view representation learning. We have now extended the Related Work section to include key references on this topic and to position our work within this recent literature. We expand on some of these points in RC5. Regarding the relation to iVAE, we refer to our answer to RC3 of Reviewer i1mE, where we draw the distinctions between iVAE and VAIR.
>
>
> **RC2: Bijectivity assumption** Following the reviewer's comments, we now address the bijectivity assumption explicitly after Definition 3, and point more clearly to the related Appendix A after Theorem 1. Following RC1.1, the abstract now states that disentanglement holds only under these assumptions, which are nonetheless standard in the VAE literature.
>
>
> **RC3: Observational vs. experimental data** We note that the sentence highlighted by the reviewer refers to a data-gathering scenario rather than to a particular architecture. Indeed, in Sec. 4.1 we train all VAE variants under this *interventional* regime (i.e. we feed the paired data $(x, y_A(x))$ to VAE and TC-VAE), showing that access to such data alone is not sufficient to match the disentanglement reached by VAIR.
>
>
> **RC4: "Proven for minAIRs" vs. "approximated by VAIR"** We agree that the sentence in the abstract could be read as claiming that VAIR provably disentangles, which we do not claim. We have rephrased it to state that VAIR approximates minAIRs, as well as adapting similar statements in Sec. 3.
>
> **RC5: Missing literature** We thank the reviewer for pointing us to this literature, some of which we were not aware of. We have extended the Related Work section accordingly.
>
> Regarding JEPA, while it shares the broad aim of constructing compact representations, it operates in a different paradigm from VAIR, as the reviewer acknowledges. We nonetheless believe that combining AIR with JEPA is an exciting line of research, and we now refer to this in the outlook.
>
> Regarding causal representation learning, we have used the provided references to explain that, crucially, we assume no causal graph and no causal relation among the factors of variation, which is a key ingredient of CRL. The data structure we consider therefore differs from typical causal datasets. We agree that spelling out these differences strengthens the work, and we now discuss them, together with the new references, in more depth.
>
> **Minor: Error bars for Sec. 4.2** We thank the reviewer for pointing this out. Fig. 3b and d indeed shows the average over 20 different models, which we now state explicitly in the main text and caption. The lower panels of Figs. 3b and 3d show representations from a single model, as they are not intended as a benchmark metric but rather as an example of a recovered representation. For Fig. 4d, we now show the average over 14 models; the result is unchanged, and the error bars are too small to be visible in the plot.

---

### Review · Reviewer_1pjk · 2026-07-02

**Summary Of Contributions:**

The paper introduces Action-Induced Representations (AIRs), a framework where different representations of the physical world are modeled given experimental actions (measurements/probes) that can be performed on them. The paper proves (Theorem 1) that factors appearing in the intersection of multiple actions' dependency sets are provably disentangled in any minimal AIR. They propose VAIR, a VAE variant with a dual-encoder architecture that can achieve disentangled representations where standard VAEs cannot. Experiments on synthetic, classical physics, and quantum systems show VAIR recovers interpretable disentangled representations, though a gap remains between the theoretical guarantees (which require exact minAIR conditions) and the practical algorithm (which only heuristically approximates them).

Strengths:

1. The paper is well-written and the experiments are well-thought-out for a framework illustration.
2. The action-induced framework is well-motivated and intuitive

Weaknesses:

1. While compelling, the experiments are very small-scale and larger evaluations on established benchmarks (e.g. dSprites, ShapeWorld, CelebA) are missing. Relatedly, the architecture uses simple 3-layer MLPs. It's unclear how VAIR would scale to high-dimensional observations (images, time series) or large action spaces.
2. There’s a gap between the theoretical considerations and the empirical results. The abstract states that VAIR can extract AIRs and therefore achieve provable disentanglement. However, Theorem 1's guarantees assume that the learned representation being a minAIR, and the author(s) acknowledge that VAIR uses a heuristic. To me, this creates a gap between the paper's claim of provable disentanglement and what the algorithm can deliver in practice.

**Audience:**

Yes

**Audience Explanation:**

Yes. Learning disentangled and interpretable representations is an active and growing area of research and the paper will be of interest to this community.

**Claims And Evidence:**

Yes

**Claims Explanation:**

The experiments are well-thought-out for a framework illustration.

**Requested Changes:**

With respect to the weaknesses, I see some critical changes along the lines of:

1. The empirical evaluations, while well-designed for illustrating the framework, are limited to low-dimensional settings and simple architectures, leaving open the question of whether VAIRs are scalable. How does VAIR perform as the number of underlying factors/actions/dimensionality grows? A systematic scaling analysis, even on synthetic data with increasing dimensionality, would substantially strengthen the paper's practical relevance.
2. More empirical evidence showing that VAIRs closely approximates minAIR conditions will strengthen the claim (though not provide provable evidence). How consistent are the disentangled representations across multiple seeds? Does ELBO consistently produce polarized outputs? If not, are some actions more impacted than others?

Questions to the authors:

1. I recognize that evaluation on standard image-based benchmarks and integration with convolutional architectures may be outside of the revision scope. However, could you comment on aspects like: what constitutes an "action" in standard benchmark settings, anticipated architectural modifications for high-dimensional observations, any expected bottlenecks? I think even a discussion of this might help with a vision for the framework beyond the current toy experiments.
2. You mention integrating VAIR into RL pipelines as future work. But RL typically assumes actions change system state, while your framework assumes fixed intrinsic properties. How do you reconcile this?
3. The polarized regime assumption (σ² → 0 or σ² → 1) is idealized. In practice, neurons exist on a continuum. How robust is the theorem to "soft" masking? And perhaps more generally, is the assumed binary structure a good world model?

Minor comments:

1. Some citations show us as (Author et al., XX) in places where they should be Author et al., XX.
2. I’m not sure how much value is added by Fig. 1 since some parts of the figure like part b) are hard-to-interpret visuals and the captions just point to the main text for details. I’d suggest either removing the figure or developing it further so that it’s interpretable on its own.

---

> ### Author Response · Authors · 2026-07-15
>
> We thank the reviewer for their thorough review and their positive assessment of the action-induced framework. Below, we respond to their comments, which we believe have helped improve the manuscript.
>
> **RC1: Larger evaluations** We thank the reviewer for this important point. Regarding benchmarking against known datasets, we note that VAIR is specifically designed for settings where actions are explicitly defined, and that standard disentanglement benchmarks do not come with predefined actions. Unfortunately, they therefore do not constitute a proper benchmark for VAIR's capabilities. Constructing action-annotated versions of standard benchmarks is nonetheless a promising direction for future research.
>
> On the other hand, we note that the capacity of $E_x$ to produce good representations in larger setups is directly tied to the capacity of VAE to do so, as their tasks are indeed analogous. On the other hand, the capacity of $E_a$ to handle high-dimensional inputs is already hinted at in Sec. 4.3, where actions are continuous 4x4 matrices.
>
> We nonetheless agree with the referee that showcasing VAIR's ability to disentangle in larger setups is key for its broad applicability. To that end, we added Appendix H, where we replicate the Abstract Experiment (Sec. 4.1) in a higher-dimensional version. We show there that VAIR consistently approximates minAIRs and obtains disentangled representations, while its closest competitor in the benchmarks (TC-VAE$_{D_a}$) is unable to do so.
>
> **RC2: Extended empirical evidence of VAIR producing minAIRs** We agree that the sentence in the abstract incorrectly attributed the provable disentanglement of AIR to the VAIR architecture. We have corrected it.
>
> We note, however, that Sec. 4.1 is intentionally directed at showing that VAIR does produce minAIRs, as highlighted by the large MIG in Fig. 2d. This experiment averages over multiple models trained with different seeds, showing that disentanglement is robust and that the ELBO is minimized in all cases (see also Fig. 6). Together with the new results in App. H, we believe this provides evidence that disentanglement by AIR is approximated in VAIRs and is robust across training initializations.
>
> **Q1.1: What constitutes an "action" in standard benchmark settings** As noted above, we work in a different paradigm from typical disentanglement benchmarks. One way these could be augmented to match the AIR paradigm would be actions whose outcomes depend only on part of the factors of variation (e.g. an image in dSprites changes color only if it contains a square). However, as stated in the text, we believe the most promising application is in experimental scenarios, where actions carry a physical meaning.
>
> **Q1.2: Anticipated architectural modifications for high-dimensional observations** Aside from the modifications one would apply in any typical VAE or ML training setup (increasing parameter count, selecting appropriate network layers), VAIR requires no particular changes. We have updated the text to note this and to point the reader to the new App. H.
>
> **Q1.3: Expected bottleneck** The main bottleneck is access to a set of actions that produce intersections and can hence yield disentanglement through AIR. This is not an architectural bottleneck, but rather a call for experimentalists and practitioners to identify the right set of actions to disentangle the factors of variation in their data. We now clarify this point just after presenting Theorem 1.
>
> **Q2: RL and fixed intrinsic properties** The connection between the AIR and RL can be thought of as follows: our observation is a given RL state $s_t$, and the new state $s_{t+1}$ after performing action $a$ is our outcome $y_A$. We are then interested in RL scenarios where environments have latent factors that are not changed by the actions, only the states are. Environments where such disentanglement is facilitated do exist, but they are most prevalent in scientific settings and can be described as scientific experiments similar to our examples.
>
> **Q3: Soft polarized regime** The theorem requires the polarized regime, and it is not clear how to circumvent this. We note that the polarized regime is a common assumption throughout the VAE literature, and never fully achieved in any numerical scenario. We also note that a binary structure is as good as a soft masked model, as the world model is given not so much by the variances $\sigma^2$, but rather by the learned neuron means $\mu$. The amount of noise injected is then related to the pressure of the latent loss on the numerical model.
>
> **Minor comments:** We thank the reviewer for spotting the typos, which we have corrected. We have also changed the caption of Fig. 1b to state clearly that it is an aid to understanding the theorem and does not work as a standalone representation of the theory. We believe it is still valuable to have the figure available while reading the theory.

---

> > ### Comment · Reviewer_1pjk · 2026-07-21
> >
> > I thank the authors for their thoughtful response and the revisions made to the manuscript. The updated abstract and the addition of Appendix H address some of my concerns.
> >
> > Regarding my RC2 (empirical evidence for minAIRs), my original request specifically asked whether the ELBO consistently produces polarized outputs. The response points to MIG scores and ELBO convergence, which demonstrate that disentanglement occurs — but does not directly answer whether σ² values cleanly polarize in practice.
> >
> > Let me explain the reasoning behind my request. The paper's empirical evidence currently shows that the conclusion of Theorem 1 holds (disentanglement occurs), but this could happen for reasons other than having found a minAIR. Reporting σ² distributions across neurons, actions, and seeds would speak to whether the preconditions hold. To illustrate what I meant by "more empirical evidence showing that VAIR approximates minAIR conditions": one could also check whether active/inactive neuron assignments are consistent across seeds, or whether reconstruction degrades when active neurons are ablated. To be clear: I'm not adding these to scope — just clarifying the type of empirical evidence that would help bridge the gap between theory and practice that reviewers have noted.
> >
> > At a minimum, σ² histograms directly address my original question and would be a lightweight addition.

---

> > > ### Author Response · Authors · 2026-07-22
> > >
> > > We thank the reviewer for their positive assessment of our response, and for the further clarification on RC2. We agree that reporting the $\sigma^2$ distributions strengthens our claims regarding the polarized regime and VAIR's approximation of minAIRs.
> > >
> > > We now include a new subsection in the Appendix, showing, for the abstract dataset, the distribution for seven exemplary models together with the average of the sorted distributions over 30 models. The sorting is required because the active and passive neurons appear at different indices in different models; what is consistent across seeds is the *number* of active neurons per action, not their position, the latter being a permutation symmetry of the latent space. The table below summarizes the averaged sorted distribution ($\log\sigma^2$, mean $\pm$ standard error of the mean (SEM) over 30 models):
> > >
> > > | Latent | Action $a_1$ | Action $a_2$ |
> > > |--------|---------------|---------------|
> > > | $z_{1}$ | -4.919 ± 0.081 | -5.515 ± 0.074 |
> > > | $z_{2}$ | -3.778 ± 0.090 | -4.591 ± 0.070 |
> > > | $z_{3}$ | -0.018 ± 0.001 | -3.675 ± 0.130 |
> > > | $z_{4}$ | -0.010 ± 0.001 | -0.085 ± 0.049 |
> > > | $z_{5}$ | -0.004 ± 0.001 | -0.014 ± 0.002 |
> > > | $z_{6}$ | 0.002 ± 0.001 | -0.006 ± 0.001 |
> > >
> > > As can be seen, for the first action only two neurons are active ($\log\sigma^2 \neq 0$), while the remaining ones collapse to the prior ($\log\sigma^2 \rightarrow 0$, i.e. $\sigma^2 \rightarrow 1$). For the second action three neurons are active, again as expected. The small standard deviations show that this polarization is sharp and reproducible across seeds.
> > > We refer to this new figure in the main text when discussing the results of Sec. 4.1:
> > > > *[...] This is in accordance with expected results for $\dim Z = 4$, $\dim Z_{I_{a_1}} = 2$ and $\dim Z_{I_{a_2}} = 3$.* This behavior is robust against dataset randomization and model initialization, as we show in \cref{fig:app_logvar_dist}.
> > >
> > > The new subsection in the appendix also relates these results to the polarized regime and minAIRs, as the reviewer suggests.
> > >
> > > **Note:** To avoid confusion between versions, we will implement these changes in the pdf version once we have the responses from all reviewers.